# CDK1 dependent phosphorylation of hTERT contributes to cancer progression

Mami Yasukawa [1,13], Yoshinari Ando [1,13], Taro Yamashita [2,13], Yoko Matsuda[3,12], Shisako Shoji [4], Masaki Suimye Morioka[5], Hideya Kawaji [5,6], Kumiko Shiozawa [7], Mitsuhiro Machitani[1], Takaya Abe[8,9], Shinji Yamada [10], Mika K. Kaneko [10], Yukinari Kato [10,11], Yasuhide Furuta[8,9], Tadashi Kondo[7], Mikako Shirouzu [4], Yoshihide Hayashizaki [6], Shuichi Kaneko [2] & Kenkichi Masutomi [1✉]

The telomerase reverse transcriptase is upregulated in the majority of human cancers and contributes directly to cell transformation. Here we report that hTERT is phosphorylated at threonine 249 during mitosis by the serine/threonine kinase CDK1. Clinicopathological analyses reveal that phosphorylation of hTERT at threonine 249 occurs more frequently in aggressive cancers. Using CRISPR/Cas9 genome editing, we introduce substitution mutations at threonine 249 in the endogenous *hTERT* locus and find that phosphorylation of threonine 249 is necessary for hTERT-mediated RNA dependent RNA polymerase (RdRP) activity but dispensable for reverse transcriptase and terminal transferase activities. Cap Analysis of Gene Expression (CAGE) demonstrates that hTERT phosphorylation at 249 regulates the expression of specific genes that are necessary for cancer cell proliferation and tumor formation. These observations indicate that phosphorylation at threonine 249 regulates hTERT RdRP and contributes to cancer progression in a telomere independent manner.

[1] Division of Cancer Stem Cell, National Cancer Center Research Institute, Tokyo 104-0045, Japan. [2] Department of Gastroenterology, Kanazawa University Graduate School of Medical Science, Kanazawa 920-8641, Japan. [3] Department of Pathology, Tokyo Metropolitan Geriatric Hospital and Institute of Gerontology, Tokyo 173-0015, Japan. [4] Laboratory for Protein Functional and Structural Biology, RIKEN Center for Biosystems Dynamics Research, Yokohama 230-0045, Japan. [5] Preventive Medicine and Applied Genomics Unit, RIKEN Center for Integrative Medical Sciences, Yokohama 230-0045, Japan. [6] RIKEN Preventive Medicine and Diagnosis Innovation Program, Wako 351-0198, Japan. [7] Division of Rare Cancer Research, National Cancer Center Research Institute, Tokyo 104-0045, Japan. [8] Animal Resource Development Unit, RIKEN Center for Life Science Technologies, Kobe 650-0047, Japan. [9] Genetic Engineering Team, RIKEN Center for Life Science Technologies, Kobe 650-0047, Japan. [10] Department of Antibody Drug Development, Tohoku University Graduate School of Medicine, Sendai 980-8575, Japan. [11] New Industry Creation Hatchery Center, Tohoku University, Sendai 980-8579, Japan. [12] Present address: Oncology Pathology, Department of Pathology and Host-Defense, Kagawa University, Kagawa 761-0793, Japan. [13] These authors contributed equally: Mami Yasukawa, Yoshinari Ando, Taro Yamashita. ✉email: kmasutom@ncc.go.jp

Recent analyses of human cancer genomes revealed that somatic mutations within the promoter of human telomerase reverse transcriptase (*hTERT*) occur at high frequency in over 50 distinct cancer types[1–6]. These common promoter mutations are associated with higher expression levels of *hTERT* and a poor prognosis[5,7–9]. In humans, experiments involving live-cell imaging techniques combined with CRISPR-Cas9 genome editing demonstrated that recruitment of hTERT and *hTERC* to telomeres occurs through dynamic interactions between telomerases and the chromosome end during S-phase[10]. Although these observations indicate that recruitment of telomerase holoenzyme to the telomere is regulated in cell cycle-dependent manner, only a small subset of hTERT forms interactions with telomeres and Cajal bodies even in S-phase[10] and the regulation and function of the majority of hTERT outside S-phase is poorly understood. In addition, cell cycle-dependent regulation of *hTERT* messenger RNA (mRNA) is observed by several groups and the highest level of *hTERT* mRNA are detected in mitotic phase[11,12].

We have previously reported that hTERT has an RNA-dependent RNA polymerase (RdRP) activity, which generates double-stranded RNAs (dsRNAs) from a single-stranded RNA not only in a primer-dependent manner, but also in a primer-independent manner[12,13]. We also detected upregulation of hTERT protein, as well as RdRP activity, in mitotic phase using several cell lines[12,14].

Here, we report that hTERT is phosphorylated in a cell cycle-dependent manner and that this phosphorylation is essential for the RdRP activity and tumor formation via regulation of target gene expression independent of hTERT-mediated elongation of telomeres.

## Results

**Mitotic-specific accumulation of hTERT.** Since it has been challenging to detect endogenous hTERT[11,15], we extensively validated available hTERT-specific antibodies against hTERT, including the mouse monoclonal antibody (mAb) (clone 10E9-2), the mouse mAb (clone 2E4-2), the sheep polyclonal Abs (pAbs) abx120550, and the rabbit mAb ab3202. Specifically, we performed validation experiments by (i) immunoprecipitation (IP) with anti-hTERT antibodies followed by immunoblotting (IB) (Fig. 1a), (ii) suppression of hTERT by small interfering RNAs (siRNAs) specific for *hTERT*[12,16] followed by IP-IB (Supplementary Fig. 1a), and (iii) recovery of telomerase activity with these antibodies as gauged by quantitative direct telomerase assay[11,17,18] (Supplementary Fig. 1b, c). In each case, we found that the 10E9-2, the 2E4-2, the abx120550 and the ab32020 identified hTERT (Fig. 1a and Supplementary Fig. 1), confirming the sensitivity and specificity of these antibodies.

We previously reported that hTERT protein localizes to mitotic spindles[14] and hTERT protein expression is enriched in mitosis[12] (Fig. 1a). We reconfirmed the enrichment of hTERT protein in mitosis using the antibodies described above[11] (Fig. 1a, two right panels). In addition, we confirmed mitotic-specific accumulation of *hTERT* at the mRNA level and RNA-dependent RNA polymerase (RdRP) activities of hTERT in a cell cycle-dependent manner by IP-RdRP assay[19] using hTERT immune complexes immunoprecipitated from cell lysates with anti-hTERT mAb (10E9-2) (Supplementary Fig. 2a, b). Furthermore, we manipulated cells in mitotic phase with nocodazole treatment or double thymidine block treatment and observed the increase of hTERT expression in both cases (Supplementary Fig. 2c). Consistent with our data, Xi et al.[11] also reported that *hTERT* expression is enriched in mitotic phase by double thymidine block treatment. These observations suggest that expression of hTERT protein is regulated in a cell cycle-dependent manner and

is not due to nocodazole treatment (through stress kinases such as p38[20]) but due to mitotic entry.

**Phosphorylation of hTERT in mitosis.** To investigate hTERT regulation in mitosis, we first treated HeLa cells with nocodazole. We confirmed that cells accumulated in mitotic phase by assessing phospho-histone H3 (Ser10) levels (Fig. 1b, lower panel). When we examined the migration of endogenous hTERT in the mitotic phase by sodium dodecyl sulfate–polyacrylamide gel electrophoresis (SDS-PAGE), we found that endogenous hTERT isolated by immunoprecipitation with anti-hTERT mAb (clone 10E9-2)[14] migrated slower than ectopically expressed FLAG-tagged hTERT (Fig. 1b, upper panel). We thus speculated that endogenous hTERT in mitotic phase is post-translationally modified. We treated hTERT immunoprecipitated with anti-hTERT mAb (clone 10E9-2) from mitotic cells with λ phosphatase and found that phosphatase treatment diminished the mobility shift of hTERT protein (Fig. 1c). This observation suggested that hTERT is phosphorylated in mitosis.

To identify the mitotic phosphorylation sites in hTERT, we ectopically expressed hTERT in HEK-293T (293T) or HeLa cells followed by treatment with nocodazole to arrest cells in mitosis. We isolated hTERT by immunoprecipitation and performed mass spectrometry (MS) analysis using liquid chromatography-tandem mass spectrometry (LC-MS/MS). We used the Mascot software package (version 2.5.1; Matrix Science) to search for the mass of each peptide ion peak against the SWISS-PROT database (Homo sapiens, 20,205 sequences in the Swiss prot_2015_09.fasta file) and to identify the phosphorylation sites. We identified four phospho-peptides in 293T cells and one phosphopeptide in HeLa cells. The phosphopeptide [241]GAAPEPERpTPVGQGSWAHPGR[261] was found in both 293T and HeLa cells (Fig. 1d). Moreover, phosphorylation sites analysis by the Mascot software revealed that only the residue at threonine 249 (T249) was phosphorylated in the phosphopeptide (amino acids 241-261) (Supplementary Data 1). We found that T249 is conserved only in primates (Fig. 1e).

To interrogate the consequences of hTERT phosphorylation at T249, we generated rabbit polyclonal antibodies that specifically recognize phosphothreonine 249 (anti-249T-P). We confirmed that treatment of hTERT isolated from mitotic HeLa cells with λ phosphatase or suppression of hTERT by hTERT-specific siRNAs ablated the signal detected by this antibody (Fig. 1f, g). Taken together, these observations indicate that hTERT at T249 is phosphorylated in mitosis.

We noted that the sequence surrounding T249 contains [S/T]P motif that is implicated as a target of the serine-threonine kinase cyclin-dependent kinase 1 (CDK1)[21]. To investigate whether CDK1 phosphorylates hTERT, we performed in vitro kinase assay using recombinant hTERT fragment proteins (amino acids 191–306) and CDK1-cyclinB proteins followed by MS analysis. We used phosphate-affinity gel electrophoresis (Phos-tag SDS-PAGE) to detect the phosphorylation status of hTERT fragment incubated with purified CDK1-cyclinB or IKK2_2-664 used as a control of serine-threonine kinase. Phos-Tag SDS-PAGE allows the migration delay of phosphorylated proteins relative to their unphosphorylated counterparts in gels containing a phosphate-binding tag[22]. The Phos-tag SDS-PAGE revealed that hTERT_191-306 proteins were phosphorylated by CDK1-cyclinB and IKK2_2-664 in vitro (Fig. 1h, upper panel). The hTERT_191-306 protein phosphorylated by CDK1-cyclinB was detected with the anti-249T-P antibodies, but we were unable to detect the protein phosphorylated by IKK2_2-664 with the anti-249T-P antibodies (Fig. 1h, lanes 6 and 7 on lower panel), suggesting that IKK2 does not phosphorylate hTERT at T249 in vitro. To further confirm that hTERT T249 is phosphorylated by CDK1-cyclinB

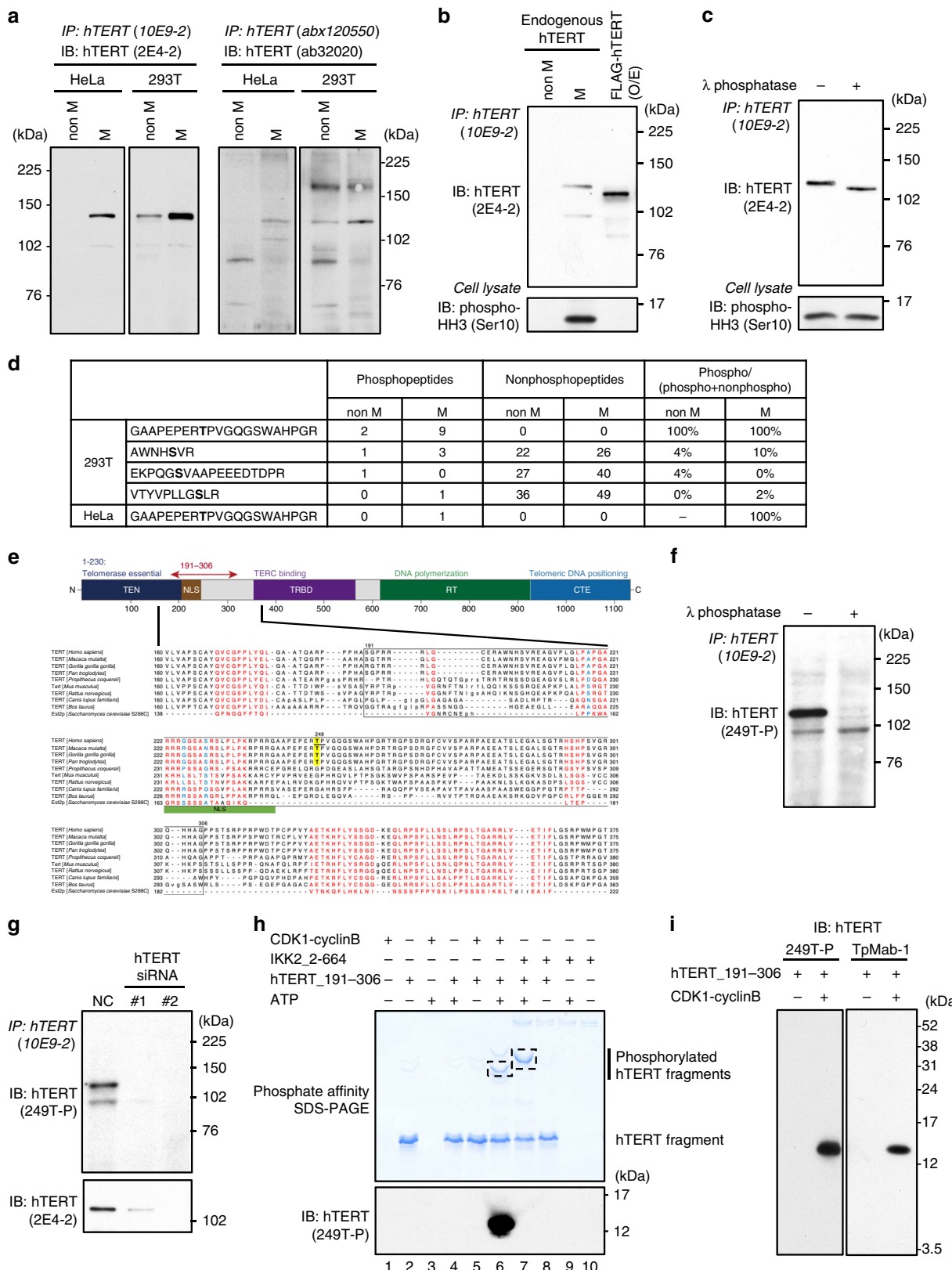

in vitro, we performed MS analysis using LC-MS/MS and identified that threonine 249 was phosphorylated by CDK1-cyclinB, not by IKK2_2-664 (Supplementary Fig. 3 and Supplementary Table 1). These observations confirmed that CDK1-cyclinB phosphorylates hTERT T249.

To further study hTERT phosphorylation at T249, we generated a mouse monoclonal antibody that specifically recognize phosphothreonine 249 (TpMab-1). We confirmed that the antibody specifically detected phosphorylation signals of

**Fig. 1 Identification of phosphorylation site of hTERT in mitosis. a** Detection of endogenous hTERT. The cells were treated with DMSO (denoted as "non M") or nocodazole to manipulate cells in mitosis (denoted as "M"), and immunoprecipitated with anti-hTERT mouse mAb (10E9-2) or anti-hTERT sheep pAbs (abx120550). The hTERT proteins were detected by anti-hTERT mouse mAb (2E4-2) (two left panels) or anti-hTERT rabbit mAb (ab32020) (two right panels). **b** hTERT proteins were isolated by immunoprecipitation with 10E9-2 and detected by 2E4-2 (upper panel). Cells arrested in mitosis with nocodazole were confirmed by anti-phospho-histone H3 (Ser10) antibodies (lower panel). **c** Endogenous hTERT immunoprecipitated from 293T cells were treated with λ phosphatase to remove the phosphate groups and detected by 2E4-2 (upper panel). **d** Summary of MS analysis to identify phosphorylation sites in hTERT. The numbers of phosphopeptides and nonphosphopeptides are based on pep_expect (Expectation value for PSM) < 0.05 used as a cutoff value. Bold letters indicate amino-acid residues identified as phosphorylated residues. See the details in Supplementary Data 1. **e** Multiple sequence alignment of telomerases using COBALT[85]. Red letters denote conserved residues and blue letters denote non-conserved residues with no gaps. Yellow is the phosphorylated threonine residue detected in this study. **f** The immune complexes were treated with λ phosphatase. Phosphorylation of hTERT at T249 was detected by anti-249T-P rabbit pAbs. **g** Endogenous hTERT proteins from cells transfected with two different siRNAs specific for *hTERT*. Phosphorylation of hTERT at T249 was detected by anti-249T-P pAbs (upper panel). Whole-hTERT proteins were detected by 2E4-2 (lower panel). **h** The recombinant hTERT fragment proteins (191–306 a.a) were phosphorylated by CDK1-cyclinB or IKK2_2-664 in vitro and phosphorylated proteins were detected in Phos-Tag SDS-PAGE (upper panel). Phosphorylation of hTERT at threonine 249 by CDK1-cyclinB was confirmed by anti-249T-P pAbs (lower panel). The protein bands enclosed with square were analyzed by MS to confirm the phosphorylation sites. **i** Phosphorylation of threonine 249 by CDK1-cyclinB was confirmed by anti-249T-P pAbs and TpMab-1 mouse mAb. Experiments were repeated three times (for **a**, **g**, **i**), five times (for **b**, **c**, **f**) and twice (for **h**) with similar results. Source data are provided in the Source Data file.

hTERT T249 phosphorylated by CDK1-cyclinB in vitro similar to anti-249T-P polyclonal antibodies (Fig. 1i).

**Phosphorylation of hTERT at T249 occurs in cancers.** To assess whether T249 was phosphorylated in human cancers, we first validated the specificity of the anti-249T-P pAbs for immuno-histochemical (IHC) staining in Huh7 xenografts. The observed signals were fully abolished by the absorption treatment with phosphopeptide (Supplementary Fig. 4a, b), whereas treatment with nonphosphopeptide did not alter the signals (Supplementary Fig. 4c), indicating that the antibodies are specific and applicable for immunohistochemical analyses.

To investigate the relationship between a progression of pancreatic cancer and hTERT phosphorylation, we stained each pancreatic precancerous lesions with anti-249T-P antibodies. We found no evidence for anti-249T-P staining in normal ducts, PanIN-1 and -2 but observed robust staining in PanIN-3 (Fig. 2a–e). In pancreatic adenocarcinomas, we found staining of hTERT phosphorylation with anti-249T-P pAbs (Fig. 2f). Both normal and atypical multipolar mitotic images exhibited strong hTERT phosphorylation (Fig. 2g, h). We also found strong staining with anti-CDK1 pAbs in pancreatic cancer cells (Fig. 2i). Furthermore, we examined the number of cells stained with anti-249T-P antibodies, referred as 249T-P-positive cells, in each pancreatic ductal lesion (Fig. 2j). The number of 249T-P-positive cells was the highest in carcinomas, followed by PanIN-3, PanIN-2, PanIN-1, and duct epithelium ($p < 0.05$). To consolidate the specificity of the data for IHC with anti-249T-P pAbs, we stained the pancreatic cancer samples with TpMab-1 monoclonal antibody, that also specifically recognize phosphothreonine 249 (Fig. 1i), in serial sections prepared from the identical pancreatic cancer lesions (Supplementary Fig. 5a–f). We observed the identical staining pattern with these antibodies in the same cells and confirmed that phosphorylation signals of hTERT T249 were detected in pancreatic carcinoma. Further, we found that 249T-P-positive cases showed higher incidence of lymph node metastasis than 249T-P-negative cases (Supplementary Table 2 and Supplementary Fig. 6a). Surprisingly, clinicopathological analysis indicated the strong correlation between CDK1 expression and hTERT T249 phosphorylation in pancreatic cancer ($p = 0.0289$, Supplementary Table 2). We confirmed that the same pancreatic cancer cells were co-stained with anti-CDK1 and TpMab-1 by immunofluorescence images (Supplementary Fig. 7a–d). The signal intensities between these antibodies demonstrated positive correlation at single-cell level ($p = 0.0350$, Supplementary Fig. 7e).

In addition, we conducted IHC with anti-249T-P pAbs and anti-CDK1 pAbs on HCC samples. We failed to detect the specific signals in non-cancerous lesions obtained from two individual patients (Fig. 2k, l) but found strong staining with anti-249T-P (Fig. 2m, n) and anti-CDK1 (Fig. 2o, p) in HCC lesions obtained from the identical patients, respectively. We found strong hTERT phosphorylation signals mostly in mitotic cells of more advanced cancers, suggesting that hTERT phosphorylation at T249 is associated with HCC grade (Supplementary Table 3 and Supplementary Fig. 6b). 249T-P-positive HCC showed high CDK1 expression with statistical significance ($p < 0.0001$, Supplementary Table 3). In addition, CDK1 expression correlated with poorly differentiated morphology (Supplementary Table 4).

In consonance with this observation, we found that patients whose tumors lacked anti-249T-P or anti-CDK1 staining had longer overall survival (OS; $p = 0.048$ and $0.036$, respectively) than patients whose cancers showed with anti-249T-P staining (Fig. 2q, r). These observations suggest that both hTERT phosphorylation at T249 and high expression of CDK1 occurs in cancers, more frequently in aggressive cancers.

**Consequences of hTERT phosphorylation.** To study the consequence of hTERT phosphorylation, we used λ phosphatase to remove all phosphate groups from hTERT immunoprecipitated with the anti-hTERT mAb (clone 10E9-2) and then performed a quantitative direct telomerase assay and an RdRP assay (Fig. 3a). We found that phosphatase treatment of hTERT reduced hTERT-dependent RdRP activity but had no measurable effect on telomerase activity.

To investigate whether endogenous hTERT protein is phosphorylated by CDK1 in vitro and the phosphorylation event regulates RdRP activity, we treated the hTERT immune complex with phosphatase to remove phosphate groups from hTERT protein and then added CDK1 and ATP to phosphorylate T249 in endogenous full-length hTERT. We confirmed that T249 residue of hTERT was phosphorylated by CDK1 using in vitro kinase assay (Fig. 3b, left panel). In addition, we noted that CDK1 treatment after phosphatase treatment did not completely relocate the hTERT signal to the original position. This indicates that there may exist other phosphorylation site(s) besides T249 by CDK1. We further confirmed that this phosphorylation event by CDK1 regulates RdRP activity of hTERT (Fig. 3b, right panel).

To confirm that CDK1 phosphorylates hTERT at T249, we suppressed CDK1 with CDK1-specific siRNAs (Fig. 3c). Suppression of CDK1 decreased cell proliferation by 70%, and we observed that introduction of CDK1-specific siRNAs #1, 2, and

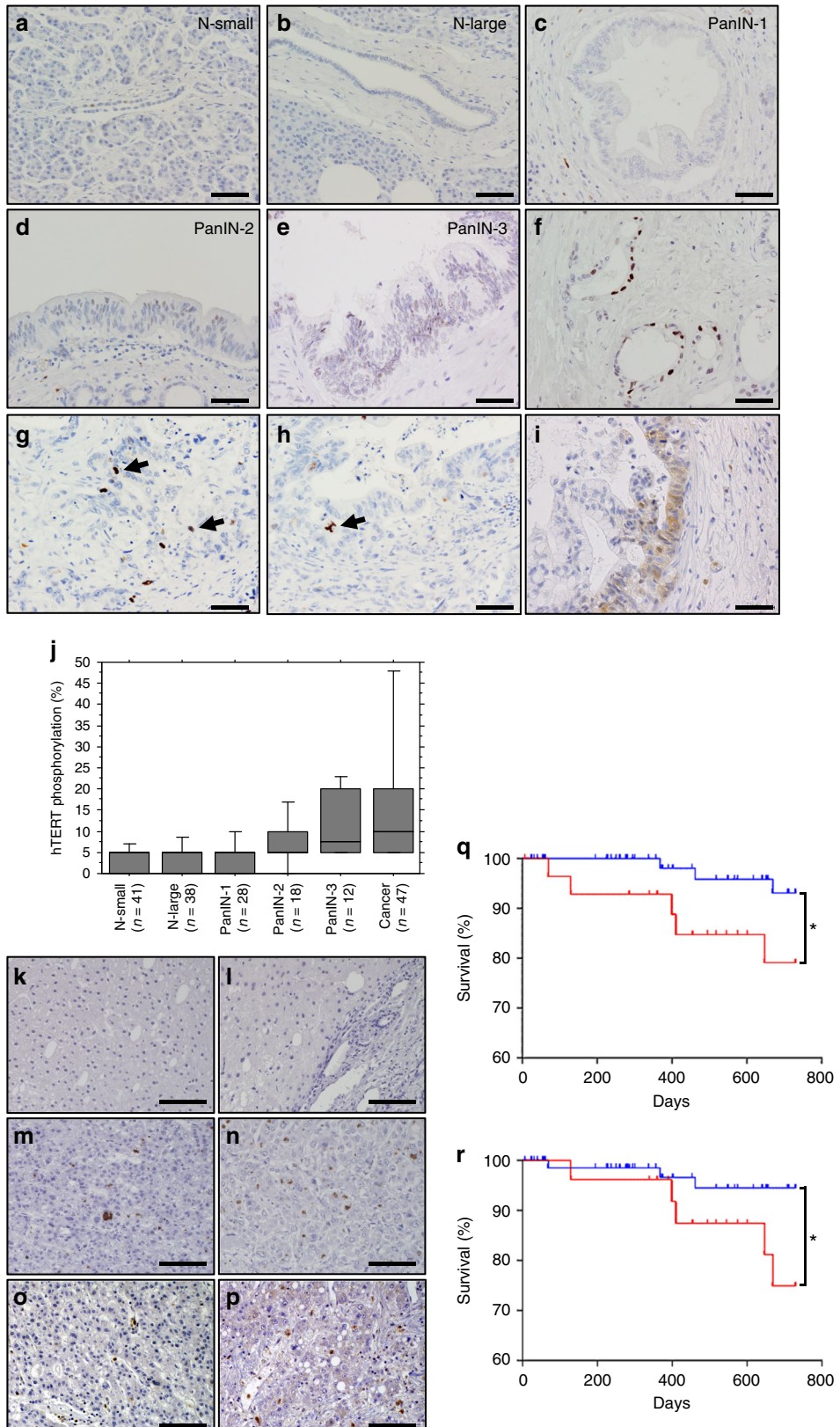

3 suppressed hTERT T249 phosphorylation by 98%. Altogether, these observations strongly suggest that CDK1 is required for the phosphorylation of hTERT at T249.

Moreover, we found that the RdRP activity of hTERT was substantially diminished by CDK1 knockdown while suppression of CDK1 had no effect on telomerase activity (Fig. 3d), indicating that phosphorylation by CDK1 is required for the RdRP activity of hTERT. These results suggest that hTERT proteins are phosphorylated by CDK1 in mitosis and the phosphorylation is not required for the telomerase activity but for the RdRP activity of hTERT.

To further confirm that CDK1 specifically phosphorylates hTERT T249, we treated HeLa cells with CDK1 inhibitor, RO-

**Fig. 2 Phosphorylation of hTERT T249 in pancreatic cancer and HCC. a–e** Representative images of precancerous lesions stained with anti-249T-P pAbs. N-small, intercalated duct to intralobular duct; N-large, interlobular duct to main pancreatic duct; PanIN, pancreatic intraepithelial neoplasia; Cancer, pancreatic invasive ductal adenocarcinoma. Original magnification x400, scale bar: 50 μm. **f** Phosphorylation of hTERT T249 was detected in the nucleus of pancreatic adenocarcinomas. Original magnification x400, scale bar: 50 μm. **g, h** Both normal mitotic (pointed by arrows, **g**) and atypical multipolar mitotic images (pointed by arrow, **h**) exhibited phosphorylation of hTERT T249 (inserts). Original magnification x400. **i** Expression of CDK1 was detected in the nucleus of pancreatic adenocarcinomas. Original magnification x400, scale bar: 50 μm. **j** Relationship between pancreatic ductal lesions and hTERT phosphorylation at T249. Box-and-whisker plot of hTERT phosphorylation for each of the pancreatic ductal lesions surgically resected from patients with pancreatic cancer ($n = 47$). The box portion of the box plot is defined by two lines at the 25th percentile and 75th percentile. A line is drawn inside the box at the median. The two whisker boundaries are the 5th percentile and 95th percentile. $p < 0.05$ by Speaman's rank correlation test. **k–p** IHC staining with anti-249T-P pAbs in non-cancerous lesions (**k, l**) and HCC lesions (**m, n**) obtained from two individual patients. IHC staining with anti-CDK1 pAbs in HCC lesions (**o, p**) obtained from two identical patients. Original magnification x200, scale bar: 100 μm. **q, r** Kaplan–Meier survival analysis of primary patients with HCC grouped for 249T-P-positive (red) or negative (blue). Phosphorylation status of hTERT T249 (**q**) and CDK1 expression (**r**) showed statistically significant correlations with overall survival (OS, $p = 0.048$ and $0.036$, respectively, log rank test, two-sided) in patients.

3306[23]. Inhibition of CDK1 activity diminished the phosphorylation of hTERT T249 (Fig. 3e) and reduced the RdRP activity of hTERT without affecting telomerase activity (Fig. 3f). Thus, both suppressing CDK1 expression or inhibiting CDK1 kinase activity decreased the phosphorylation of hTERT at T249 and hTERT-mediated RdRP activity.

**Consequences of phosphorylation of hTERT on T249**. To further assess the consequences of phosphorylation of hTERT T249 in the human cell lines, we created hTERT mutants where we substituted the threonine 249 of hTERT with alanine (T249A) or glutamic acid (T249E). We introduced these mutants in telomerase-negative human fibroblast BJ cells and analyzed the telomerase activity of the mutant proteins by the telomeric repeat amplification protocol (TRAP) assay (Fig. 4a, left panel and Supplementary Fig. 8a) and direct telomerase assay (Supplementary Fig. 8b). We found that both of these mutant hTERT alleles exhibited telomerase activity equivalent to that observed for wild-type hTERT (WT-hTERT). In contrast, when we examined RdRP activity using BJ cells stably overexpressing these hTERT mutants, BJ cells with WT-hTERT and T249A exhibited little RdRP activity while T249E cells demonstrated steady state level of RdRP activity. These data indicate that normal cells show little RdRP activity due to the lack of phosphorylation at T249 of hTERT (Supplementary Fig. 8c).

We confirmed the findings by introducing these hTERT mutants in 293T cells and measured the telomerase activity by a quantitative assay as measured by the direct telomerase assay (Fig. 4b). We verified that these mutants retained telomerase activity comparable to WT-hTERT and did not interfere with endogenous hTERT. Consistent with a previous report[11], ectopic expression of WT-hTERT protein slightly increased measured telomerase activity. We also assessed whether these two mutants elongated telomeres when expressed in cells by performing Southern blotting for telomere restriction fragments. We analyzed the telomere length in the BJ cells at 8 population doubling (PD) after infection of the two mutants and found that T249A and T249E mutants elongated telomeres (Fig. 4a, right panel). These data demonstrated that telomerase activity of the mutants was intact.

We next examined whether introduction of hTERT mutants in 293T cells altered hTERT-dependent RdRP activity[13]. We found that expression of the phosphorylation-defective mutant hTERT T249A decreased RdRP activity (Fig. 4c), while ectopic expression of hTERT T249E increased RdRP activity, suggesting that this mutant acts as a phosphomimetic mutant. These observations confirmed that the phosphorylation of hTERT T249 is required for the RdRP activity but not for reverse transcriptase activity of hTERT.

To further assess the biological effects of phosphorylation of hTERT T249 in cancer cell lines, we ectopically expressed hTERT T249A, hTERT T249E, or a control vector expressing only a drug resistance marker into several well-characterized cancer cell lines; two telomerase/RdRP-positive ovarian cancer cell lines[12], PEO1 and PEO14, and a telomerase/RdRP-positive patient-derived hepatocellular carcinoma (HCC) MT cells (Fig. 4d). In PEO1, PEO14, and MT cells, the expression of hTERT T249A dramatically decreased the cell proliferation (0.73-, 0.18- and 0.50-fold, respectively) while the hTERT T249E showed a minor effect on cell proliferation (1.03-, 1.01- and 1.19-fold, respectively) (Fig. 4e). These observations suggest that phosphorylation-defective hTERT T249A interferes with hTERT-mediated RdRP activity and the inhibition of RdRP activity causes inhibitory effects on the cell growth. In contrast, hTERT T249E failed to alter the proliferation of these cell lines.

To verify the effects of hTERT mutants on the cell proliferation in an hTERT null cell line, we introduced WT-hTERT, hTERT T249A, hTERT T249E, or a control vector in Saos2 cells that lack telomerase activity. We observed that T249E mutant most effectively accelerates the cell proliferation, WT-hTERT is the second and that T249A has no effects on cell proliferation (Supplementary Fig. 9).

To examine whether suppression of hTERT affects the cell viability, we introduced hTERT-specific siRNAs to several different cancer cell lines known to be hTERT positive (A549, 293T, and HeLa cells) and an hTERT null cell line (VA13). We confirmed that cancer cells addicted to TERT rapidly die by suppression of hTERT (Supplementary Fig. 10).

To further evaluate the influence of hTERT phosphorylation at T249 using patient-derived xenograft (PDX) models in vivo, we introduced hTERT T249A (MT+T249A) or a control retroviral vector (MT-Control) into MT (patient-derived HCC) cells and implanted these cells subcutaneously into non-obese diabetic/severe combined immunodeficiency (NOD/SCID) mice. Six of eight mice harboring MT+T249A failed to form tumors (Fig. 4f, left panel) and the volume of tumors from mice harboring MT+T249A was significantly smaller than those from mice with MT-Control (Fig. 4f, right panel; $p = 0.0028$). Taken together, these observations indicate that phosphorylation of hTERT at T249 is required for cancer cell proliferation through activation of RdRP activity of hTERT. Although the expression of these hTERT alleles clearly separated hTERT-mediated telomerase and RdRP activity, endogenous hTERT is expressed at low levels even in cancer cells.

**CRISPR-Cas9 genome editing to introduce mutations at T249.** To assess whether the phosphorylation of T249 affected endogenous hTERT function, we used CRISPR-Cas9 genome editing to introduce alanine or glutamic acid substitutions at T249 in the

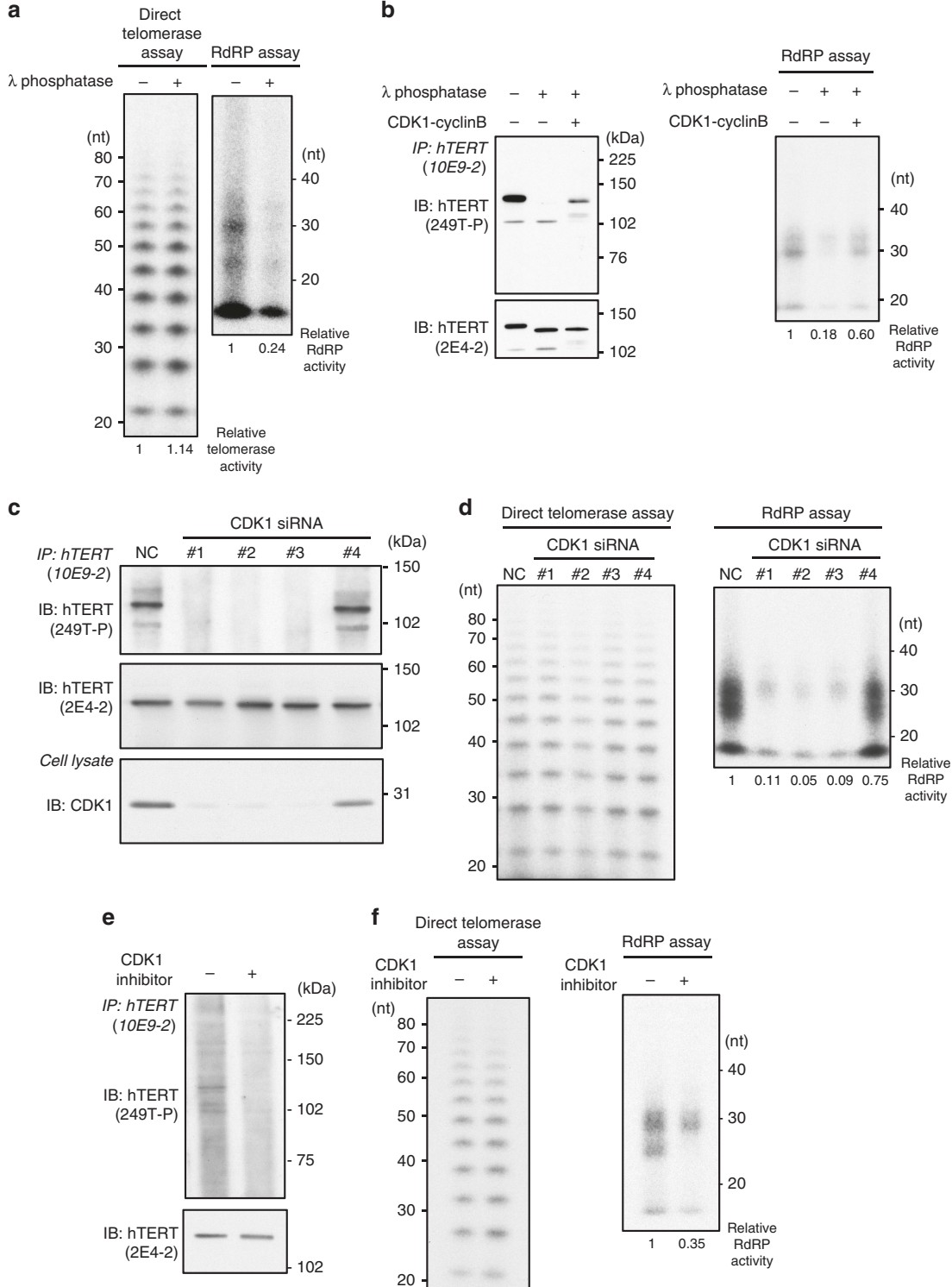

endogenous *hTERT* locus in 293T cells (Fig. 5a). After intro-duction of Cas9 and guide RNA with donor DNAs into 293T cells, we selected clones and confirmed that the mutations were introduced by PCR amplification, restriction enzyme digestion, and Sanger sequencing (Fig. 5b, c). Control cells, Control-CRISPR, were selected from clonal selection of cells expressing Cas9 and no guide RNAs. We examined potential off-target (OT) mutations in six candidate loci corresponding to guide RNA sequence and no OT mutations were detected in the cells (Supplementary Fig. 11). We confirmed that T249A-CRISPR

and T249E-CRISPR were expressed at a level similar to that of endogenous hTERT in the Control-CRISPR cells (Fig. 5d, lower panel). We further ascertained that introduction of these muta-tions into hTERT had no effect on the protein stability, reported protein-protein interactions with hTERT, such as nucleostemin (NS)[16] and SMG6[24–26], cellular localization, and a template-independent terminal transferase activity known as a non-canonical activity of hTERT[27] (Supplementary Fig. 12a–d).

Using these genome-edited 293T cells, we confirmed that substitution of T249 to alanine ablated hTERT-dependent RdRP

**Fig. 3 CDK1 phosphorylates hTERT. a** Direct telomerase assay (left panel) and IP-RdRP assay (right panel) using HeLa cells. Immune complexes were treated with λ phosphatase to remove the phosphate groups from hTERT proteins. The relative activities of telomerase and RdRP are noted below the panel, respectively. **b** In vitro CDK1 kinase assay of hTERT immune complexes treated with λ phosphatase, followed by IP-IB assay (left panel) and IP-RdRP assay (right panel). The relative RdRP activities are noted below the panel. **c** Detection of endogenous hTERT proteins using HeLa cells transfected with four different siRNAs specific for *CDK1* followed by nocodazole treatment. Phosphorylation of hTERT proteins were detected by anti-249T-P pAbs (upper panel). Suppression of CDK1 proteins was confirmed by anti-CDK1 mAb (lower panel). **d** Direct telomerase assay (left panel) and IP-RdRP assay (right panel) using HeLa cells transfected with four different siRNAs specific for *CDK1*. **e** Detection of phosphorylated hTERT proteins using HeLa cells treated with 5 μM of CDK1 inhibitor, RO-3306. The cells were then treated with nocodazole. Phosphorylation of hTERT proteins was detected by anti-249T-P pAbs (upper panel) and hTERT proteins isolated by immunoprecipitation were detected by anti-hTERT mAb (clone 2E4-2) (lower panel). **f** Direct telomerase assay (left panel) and IP-RdRP assay (right panel) using HeLa cells treated with 5 μM of CDK1 inhibitor, RO-3306. The relative RdRP activities are noted below the panel. Experiments were repeated three times (for **a** (right panel), **c**, **e**, **f** (right panel)) and twice (for **a** (left panel), **b**, **d**, **f** (left panel)) with similar results. Source data are provided in the Source Data file.

activity (Fig. 5d, upper panel). Similar to ectopic expression of the mutant proteins (Fig. 4c), hTERT with T249E mutation increased the RdRP activity while hTERT with T249A mutation decreased the activity. These results confirmed that phosphorylation of T249 is essential for RdRP activity of hTERT.

To evaluate the direct influence of hTERT phosphorylation at T249 using xenograft models in vivo, we inoculated Control-CRISPR or T249A-CRISPR subcutaneously into NOD/SCID mice. Four of eight mice harboring T249A-CRISPR failed to form tumors (Fig. 5e, upper panel) and the volume of tumors from mice with T249A-CRISPR was significantly smaller than those from mice with Control-CRISPR (Fig. 5e, lower panel; $p = 0.0469$). We confirmed that RdRP-deficient hTERT proteins isolated from T249A-CRISPR retained telomerase activity and bound to *hTERC* RNA comparable to the hTERT from Control-CRISPR (Fig. 5f). Taken together, these observations favor that phosphothreonine 249 in hTERT is important for cancer cell proliferation through activation of RdRP activity of hTERT.

**Effects of hTERT phosphorylation on gene expression.** In prior work, we showed that hTERT-mediated RdRP activity affects gene expression by affecting RNA levels[14]. Global, low-level transcription are retained during mitosis[28]. We hypothesized that alteration of hTERT RdRP activity would change transcription regulatory network to trigger proliferation of cancer cells. To assess the impact on transcriptome by hTERT phosphorylation at T249, we examined gene expression at promoter level using Cap Analysis of Gene Expression (CAGE) method[29] for T249A-CRISPR phosphorylation deficient cells, by contrasting to Control-CRISPR cells in triplicate. We found 3,482 CAGE peaks, indicating each transcriptional start site (TSS) and the expression level as a promoter activity, that exhibited altered the activity in these cells (FDR < 1e-4), where 1,926 and 1,556 were up- and down-regulated in T249A-CRISPR. In particular, we found that three independent CAGE peaks of the transcription factor, forkhead box O4 (*FOXO4*), were upregulated in T249A-CRISPR while the most upstream TSS, p1, showed no change (Fig. 6a). We confirmed the increase in expression of FOXO4 proteins in T249A-CRISPR by immunoblotting (Fig. 6b). We further monitored *FOXO4* mRNA level by manipulating cells in mitosis and confirmed that *FOXO4* expression level is lower in mitosis than in other cell cycle phases in Control-CRISPR cells, while we failed to observe the differences in T249A-CRISPR, which defects in RdRP activity. This finding favors that CDK1 activity and upregulated hTERT-RdRP activity may downregulates *FOXO4* expression in mitosis (Supplementary Fig. 13). To further confirm that the phosphorylation of hTERT T249 by CDK1 specifically affects the expression of *FOXO4*, we treated HeLa cells with CDK1 inhibitor, RO-3306. Inhibition of CDK1 activity increased the expression of *FOXO4* mRNA in a dose-dependent manner (Supplementary Fig. 14).

Gene ontology (GO) term analysis showed enrichment of genes associated with "cell cycle phase transition" (GO:0044770, Supplementary Table 5) within the upregulated genes in the T249A-CRISPR. Notably, *FOXO4* is annotated in this GO term and functions as tumor suppressor proteins in various cancers by preventing proper cell cycle regulation[30–35]. These findings indicate that phosphorylation of T249 on hTERT regulates the expression of *FOXO4* negatively in cancer cell lines and increase of *FOXO4* expression in T249A-CRISPR is correlated with a decrease of tumor formation.

To determine whether *FOXO4* is a direct target of hTERT phosphorylated at T249, we performed IP-RT-PCR experiments with RNAs isolated from hTERT-IP samples of Control-CRISPR and T249A-CRISPR. We isolated hTERT-RNA complexes by immunoprecipitation with an anti-hTERT mAb (clone 10E9-2) from Control-CRISPR and T249A-CRISPR cells, and purified RNAs from the immune complexes with DNase treatment. cDNAs were synthesized with strand-specific primers, and we performed PCR using primers to amplify the first exon of *FOXO4* mRNAs. We confirmed that endogenous hTERT immunoprecipitated from Control-CRISPR interacts with the first exon of *FOXO4* mRNAs and that there was a 50% reduction of hTERT associated with *FOXO4* mRNAs isolated from T249A-CRISPR cells (Fig. 6c). We also found that hTERT from Control-CRISPR interacts with antisense RNAs (asRNAs) of *FOXO4* mRNA under condition in which we failed to recover *RNase P* RNA[13]. To rule out the non-specific association between hTERT and *FOXO4* RNAs, we performed UV crosslinking followed by conventional immunoprecipitation with anti-hTERT antibody. Moreover, we washed the immune complex under stringent conditions such as high-salt condition. We confirmed that the interaction in Control-CRISPR is maintained even in such a stringent condition (Supplementary Fig. 15).

To further assess the function of asRNAs in regulation of *FOXO4* mRNA, we synthesized asRNA complementary to the first exon of *FOXO4* mRNA and transfected it in T249A-CRISPR and Control-CRISPR. We confirmed that introduction of *FOXO4* asRNA downregulates *FOXO4* mRNA in T249A-CRISPR (Fig. 6d).

To examine whether CDK1 directly regulates FOXO4 protein, we treated T249A-CRISPR phosphorylation deficient cells with CDK1 inhibitor (RO-3306). We found that there are no differences of FOXO4 protein expression level and the FOXO4 protein migrates at the similar molecular weight (Supplementary Fig. 16). These data suggest that CDK1 might not involve in FOXO4 protein regulation by a direct phosphorylation of FOXO4 protein.

Taken together, we propose a model for regulation of *FOXO4* expression via phosphorylation of hTERT at T249 (Supplementary Fig. 17). Specifically, in Control-CRISPR, hTERT proteins phosphorylated at T249 bind to the first exon of *FOXO4* mRNAs.

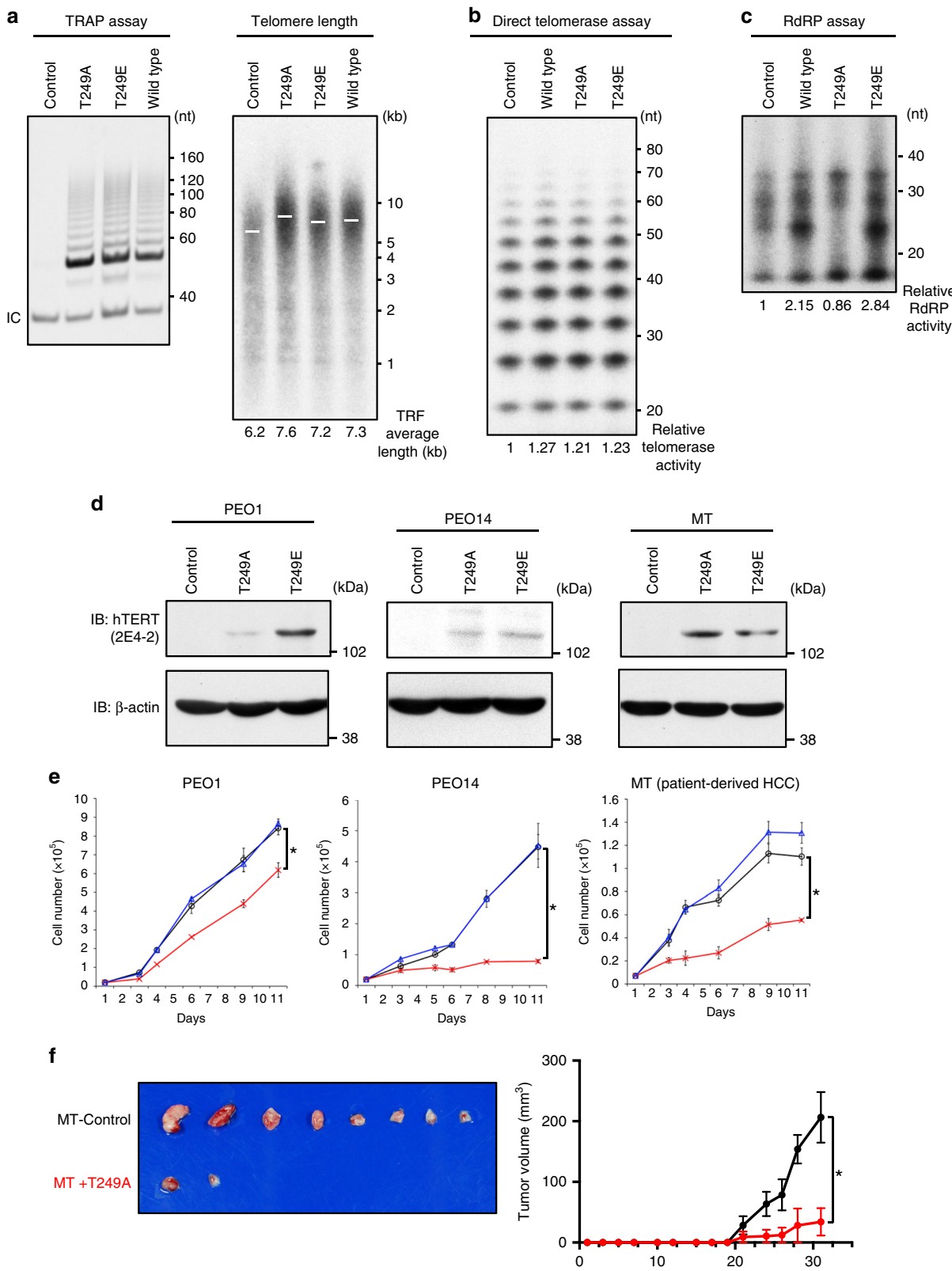

These RNAs serve as templates for dsRNA synthesis by hTERT-RdRP. These dsRNA complexes are degraded and the expression of *FOXO4* is reduced. In T249A-CRISPR, RdRP-deficient hTERT-T249A proteins interact with less *FOXO4* mRNAs than hTERT proteins in Control-CRISPR. Without producing asRNA, more *FOXO4* mRNAs are retained in the T249A-CRISPR and

increase of *FOXO4* expression causes tumor suppression via cancer cell growth arrest.

To corroborate the effects of *FOXO4* expression in cancer cells, we established stable cell lines introduced with pBABE-hygro (pBh) control vector or *FOXO4*-expressing retroviral vector (pBh-FOXO4) in HeLa cells and 293T cells. Increase of FOXO4 protein

**Fig. 4 Consequences of hTERT phosphorylation on T249. a** TRAP assay using BJ cells stably introduced with control retrovirus, hTERT mutants or wild-type hTERT (left panel). hTERT mutants were created by substitution of the threonine 249 with alanine (T249A) or glutamic acid (T249E). Telomere length in the BJ cells was determined by Southern blot analysis for telomere restriction fragments (right panel). White bars indicate the position of mean radioactivity and the TRF average length were noted below the panel. Telomere structure in BJ cells was analyzed at 8 PD after infection. **b** Direct telomerase assay using hTERT proteins immunoprecipitated from 293T cells transfected with control vector, wild-type hTERT, T249A, or T249E mutants. The relative telomerase activities are noted below the panel. **c** IP-RdRP assay using the cells prepared with identical manipulations as used in the **b**. The relative RdRP activities are noted below the panel. **d** Immunoblotting to confirm ectopically overexpressed hTERT mutant proteins in PEO1, PEO14 and MT cells infected with hTERT-T249A, hTERT-T249E, or control retroviruses. hTERT proteins were detected by anti-hTERT mAb (clone 2E4-2) (upper panel). β-actin was used as an internal control (lower panel). **e** Cell proliferation assay using PEO1, PEO14 and MT cells infected with hTERT-T249A (red line), hTERT-T249E (blue line) or control retrovirus (black line). This assay was done in triplicate and data are shown as the mean ± SD. Asterisk: statistically significant values ($p < 0.05$ by Student's $t$-test, two-sided). **f** MT cells introduced a control retroviral vector (MT-Control) or hTERT-T249A (MT+T249A) were implanted into eight NOD/SCID mice for each and tumors generated from the cells are presented (left panel). The volume of tumors from mice harboring MT-Control (black line) or MT+T249A (red line) is shown as the mean ± SEM ($n = 8$, $p = 0.0028$ by unpaired $t$-test, two-sided) (right panel). Experiments were repeated three times (for **c**, **d**) and twice (for **a**, **b**) with similar results. Source data are provided in the Source Data file.

expression introduced with pBh-FOXO4 (HeLa+pBh-FOXO4) was confirmed by immunoblotting whereas expression level of FOXO4 introduced with pBh (HeLa+pBh) was below the detection limit of anti-FOXO4 antibody (Fig. 6e, left panel). Upregulation of FOXO4 significantly inhibited the cell proliferation in vitro by 77.0% (Fig. 6e, right panel). To evaluate the effects of FOXO4 overexpression in HeLa using xenograft models, we inoculated HeLa+pBh or HeLa+pBh-FOXO4 subcutaneously into NOD/SCID mice. Retroviral expression of FOXO4 resulted in reduction of tumor formation in five of eight mice (Fig. 6f, upper panel). The volume of tumors from mice harboring HeLa+pBh-FOXO4 was significantly smaller than those from mice with HeLa+pBh (Fig. 6f, lower panel, $p = 0.0092$). Ectopic expression of FOXO4 in Control-CRISPR 293T cells also showed the same trend to delay cell proliferation and reduce tumor formation (Supplementary Fig. 18a, b). Taken together, these observations confirmed that increase of FOXO4 expression in T249A-CRISPR brings about a decrease of tumor formation in vivo.

To further assess the biological influences of increase of FOXO4 expression in T249A-CRISPR, we established stable cell lines expressing short-hairpin RNAs (shRNAs) specific for FOXO4 (shFOXO4-1, 2) or shGFP control in T249A-CRISPR. Knockdown of FOXO4 protein expression in the cell lines was confirmed by immunoblotting (Fig. 6g, upper panel). Down-regulation of FOXO4 in T249A-CRISPR significantly increased the cell proliferation by 1.31- and 1.26-fold, respectively (Fig. 6g, lower panel). To evaluate the effects of FOXO4 suppression in T249A-CRISPR on tumor formation, we used xenograft models by the subcutaneous inoculation of T249A-CRISPR expressing shFOXO4 (T249A-CRISPR+shFOXO4-1, 2) or shGFP (T249A-CRISPR+shGFP) into NOD/SCID mice. While five of seven mice with T249A-CRISPR+shGFP formed xenograft tumors, T249A-CRISPR+shFOXO4-1 and -2 developed tumors in all seven mice (Fig. 6h, upper panel). Compared to T249A-CRISPR+shGFP and Control-CRISPR, T249A-CRISPR+shFOXO4 exhibited a higher tumor growth rate and the volume of tumors from mice with T249A-CRISPR+shFOXO4-1 and -2 was significantly larger than those from mice with T249A-CRISPR + shGFP (Fig. 6h, lower panel; $p = 0.0008$ and $p = 0.0136$, respectively). In Control-CRISPR cells, stable knockdown of FOXO4 by expressing short-hairpin RNAs (shRNAs) specific for FOXO4 (shFOXO4-1, 2) did not show any statistically significant differences on cell proliferation and tumor formation (Supplementary Fig. 19a, b). These observations verified that perturbation of FOXO4 expression by specific shRNAs rescued the defects in tumor formation ability in T249A-CRISPR.

We previously reported that hTERT proteins interact with the RNA component of mitochondrial RNA processing endoribonuclease (RMRP) and produce dsRNA by RdRP activity[13]. RMRP

RNA is a non-coding RNA and mainly transcribed by RNA polymerase III[36]. We confirmed that the interaction between hTERT and RMRP was increased in mitotic cells in 293T and HeLa cells (Supplementary Fig. 20a). We also found that hTERT complexes from T249E-CRISPR bind to both sense and antisense RMRP RNAs while hTERT complexes from T249A-CRISPR fail to interact with the RNAs (Supplementary Fig. 20b). While we failed to observe the phosphorylation of hTERT causes any effects on the protein stability, protein-protein interactions, cellular localization and the known enzymatic activities (Supplementary Fig. 12a–d and Fig. 5f), we confirmed that the regulation of target RNAs expression occurs due to the alteration of the binding affinity between hTERT and endogenous template RNAs, such as FOXO4 and RMRP, and the production of antisense RNAs. These findings indicate that phosphorylation affects the binding affinity between hTERT and template RNAs for RdRP activity.

## Discussion

Telomerase plays an essential role in maintaining telomeres. The expression of the catalytic subunit of telomerase hTERT is upregulated in the majority of human cancers, most often due to mutations in its promoter. Here, we demonstrate that hTERT is phosphorylated by CDK1 in mitosis and that this phosphorylation event affects a second hTERT function, RNA-dependent RNA polymerase activity without affecting hTERT function at telomeres. This mitotic function of hTERT correlates with tumor aggressiveness and is necessary for tumor formation. Altogether these observations separate the telomere and non-telomere directed functions of hTERT and identify a mechanism that regulates hTERT RdRP activity.

In this study, we focused on the characterization of T249 phosphorylation of hTERT in pancreatic cancer and HCC, since these cancers are most deadly cancers with poor survival outcome[37]. In addition, both cancers probably occur in correlation with functional role of hTERT protein[38,39].

It is well accepted that telomerase recruitment to telomeres occurs in S-phase in many organisms[40,41]. While these observations indicate that telomerase recruitment to telomeres and telomerase activation are regulated in a cell cycle-dependent manner, there have been two reports that studied telomerase activity outside the S-phase[42,43]. Specifically, these reports showed that Xenopus telomerase activity was active in mitotic phase and modulated chromatin configuration. In consonance with these findings, we previously reported that hTERT localizes to both mitotic spindles and centromeres, that hTERT protein expression level is enriched in mitosis, and that hTERT-RdRP activity is present in nuclear extracts derived from several cancer cell lines arrested in mitosis[12,14,19]. Here, we have confirmed and extended

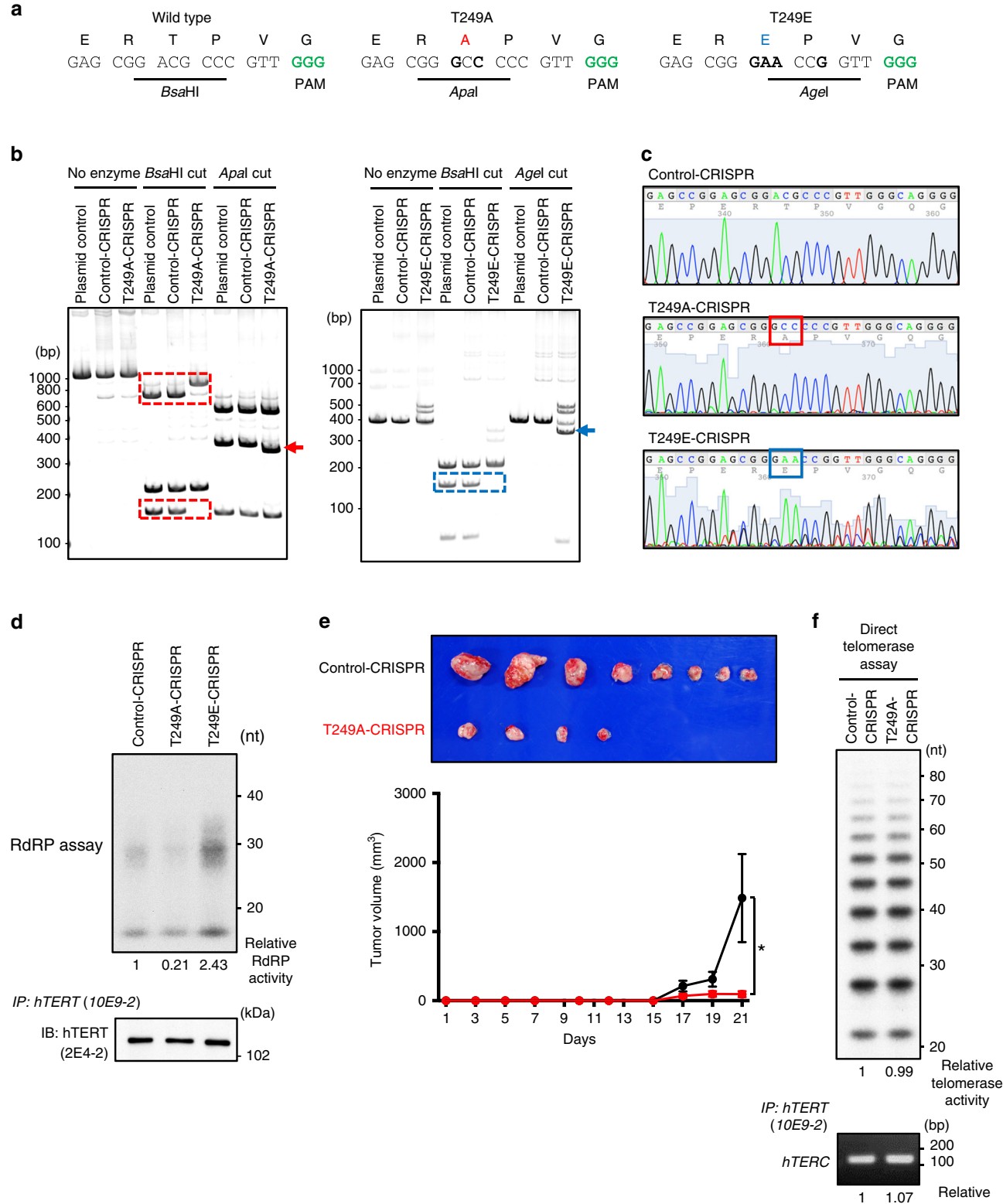

these findings by identifying the mitotic kinase CDK1 as a key regulator of hTERT RdRP but not telomerase activity.

Several laboratories have reported that hTERT is phosphorylated[44–47]. Specifically, four putative phosphorylation sites have been reported, serine 227[46], serine 457[47], tyrosine 707[45], and serine 824[44]. Serine 227 is located in a canonical nuclear localization signal, might be phosphorylated by Akt kinase, and the phosphorylation of S227 may promote nuclear localization of hTERT indicated by the experiments using phosphorylation mutant proteins[46]. Serine 824 also might be phosphorylated and the phosphorylation of S824 is suggested to enhance hTERT-mediated telomerase activity via Akt activation[44]. This previous report did not provide any direct evidence that hTERT S824 phosphorylation itself is really essential and prerequisite for the maintenance of basal telomerase activity. We speculated that although hTERT S824 phosphorylation may enhance the

**Fig. 5 Threonine 249 in the endogenous *hTERT* locus is edited by CRISPR-Cas9 in 293T cells. a** Genome editing strategy to introduce alanine or glutamic acid substitution in the endogenous *hTERT* locus. Green letters denote the PAM (protospacer adjacent motif) sequence for genome editing by CRISPR-Cas9. **b** PCR fragments surrounding target regions (1,184 bp for T249A and 456 bp for T249E) were amplified from genomic DNA of genome-edited 293T clones. PCR fragments from pBABE-puro-hTERT (plasmid control), 293T control cells (Control-CRISPR) and mutant cells (T249A-CRISPR, T249E-CRISPR) were digested with the indicated restriction enzymes and electrophoresed in 5% polyacrylamide gels. The bands enclosed with square and indicated by arrows show that the cleavage patterns of PCR fragment were changed in CRISPR-mutant clones. **c** Sanger sequencing traces of a control and two mutant clones generated from gel-purified PCR fragments of the genomic DNAs of the respective clones. The sequences boxed in red and blue are the base triplet coding for alanine and glutamic acid in mutant clones, respectively. **d** IP-RdRP assay using hTERT proteins immunoprecipitated from 293T CRISPR clones (upper panel). The relative RdRP activities are noted below the panel. hTERT proteins isolated by immunoprecipitation from the cells were detected by anti-hTERT mAb (clone 2E4-2) (lower panel). **e** Tumors generated from $1 \times 10^5$ cells of 293T Control-CRISPR or 293T-T249A-CRISPR in eight NOD/SCID mice for each are presented (upper panel). The volume of tumors from mice with Control-CRISPR (black line) or T249A-CRISPR (red line) is shown as the mean ± SEM ($n = 8$, $p = 0.0469$ by unpaired $t$-test, two-sided) (lower panel). **f** Direct telomerase assay using hTERT proteins immunoprecipitated from Control-CRISPR or T249A-CRISPR. The relative telomerase activities are noted below the panel (upper panel). Immune complexes were isolated from Control-CRISPR and T249A-CRISPR with anti-hTERT mAb (clone 10E9-2) and associated *hTERC* RNA were subjected to RT-PCR (lower panel). Experiments were repeated five times (for **d**, **f**) with similar results. Source data are provided in the Source Data file.

telomerase activity, phosphatase treatment to remove phosphate group(s) in hTERT did not affect the minimal basal activity of telomerase. Serine 457 is phosphorylated by DYRK2 during G2/M and promotes ubiquitination of hTERT proteins[47]. Tyrosine 707 might be phosphorylated by C-Src under oxidative stress and the phosphorylation of Y707 is suggested to alter subcellular localization of hTERT, shown by phosphorylation mutant protein replacing tyrosine 707 with nonphosphorylatable phenylalanine[45]. We were unable to detect these previously reported phosphorylation sites by MS analysis in this study with mitotic cells. These putative phosphorylation sites were predicted from consensus sequences for specific kinases, and MS analysis was not conducted in these previous reports (Supplementary Table 6). We speculate that these sites may be difficult to detect using MS analysis since the phosphorylation occurs only under specific conditions such as Akt overexpression or under oxidative stress. In addition, previous reports demonstrated that phosphorylation affect either telomerase activity or subcellular localization of hTERT using phosphorylation mutant proteins whereas our study identifies another phosphorylated site that does not affect telomerase activity (Figs. 4b and 5f).

We have identified CDK1 as a kinase that phosphorylates hTERT at T249 in mitosis. CDK1 is essential for cell division in the embryo and deficiency in CDK1 in mouse model results in embryonic lethality in the first cell divisions[48]. Moreover, CDK1 is sufficient to drive the cell cycle in all cell types[48]. Previous reports demonstrated that CDK1 expression is upregulated in many human malignant tumor tissues and that CDK1 activity correlates with the prognosis of patients with tumors[49–57]. In addition, loss of CDK1 in the liver in CDK1 conditional knockout mice showed complete resistance against tumorigenesis[58], indicating that CDK1 is required for tumorigenesis in liver cancer. Here, we report a link between CDK1 and phosphorylation of hTERT at T249, and the phosphorylation occurs more frequently in aggressive and advanced cancers in clinical samples, suggesting an additional role for CDK1 in cancer progression. These observations implicate CDK1 and phosphorylation of hTERT T249 as an approach to inhibit a key cancer-associated function of hTERT.

CDK1 is essential for progression into mitotic phase and is often overexpressed in human cancers[59]. Moreover, a previous report indicated that CDK1 overexpression correlated with poor clinicopathological features and survival in HCC[56], consistent with our current study. Therefore, CDK1 is considered as a pivotal molecular target. Indeed, following the success of a pan-CDK inhibitor, palbociclib, in advanced breast cancer, two phase II clinical trials (milciclib and palbociclib) are currently ongoing to evaluate their efficiency on advanced HCC. Unfortunately, milciclib targets CDK2/4/5/7 and palbociclib targets CDK4/6, and

no pan-CDK inhibitors that also targets CDK1 is currently evaluated in clinical trials[60]. Since our data demonstrated the alternative role of CDK1 to phosphorylate hTERT T249 for tumorigenesis, selective CDK1 inhibitors may have the profound impact on the treatment of aggressive cancers. Pan-CDK inhibitor(s) that also targets CDK1 would be a better candidate for treating these cancers.

Threonine 249 in hTERT is conserved in human and primates but not in the other mammals and yeast (Fig. 1e), suggesting that phosphorylation-dependent RdRP activity of TERT is restricted to primates. When we introduced T249A substitution into endogenous hTERT, we found that the mutant did not affect telomerase activity at telomeres but instead affected RdRP activity and inhibited the tumorigenic potential of human cancer cell lines. These observations suggest that phosphorylation of hTERT at T249 contributes to tumor formation independent of its reverse transcriptase activity to elongate telomeres. Furthermore, we found that phosphorylation of hTERT at T249 regulates the expression of the tumor suppressor gene, *FOXO4*, in cancer cell lines (Supplementary Fig. 17).

Expression levels of *FOXO4* inversely correlate with tumor formation and incidence of clinical metastasis[30–35]. Consistent with *FOXO4* providing a role of tumor suppressor, T249A-CRISPR expressing higher levels of *FOXO4* formed less and smaller tumors in mouse xenograft model (Figs. 5e and 6h). Our findings indicate that phosphorylation of hTERT T249 regulates *FOXO4* expression negatively by the RdRP activity in cancer cells and is implicated in pivotal event for carcinogenesis. We note that FOXO4 expression is very low and difficult to detect in endogenous protein level. In the current study, while we monitored *FOXO4* expression level as mainly gauged by mRNA, the mRNA level does not necessarily reflect the FOXO4 protein expression level nor FOXO4 functional aspect. In case of FOXO1 known as a putative tumor suppressor, phosphorylation of FOXO1 by CDK1 inhibits the transcriptional activity of FOXO1 and promotes cell proliferation[61] while CDK1 phosphorylation of FOXO1 stimulates FOXO1-dependent transcription and leads to cell death in neurons[62]. Along the same line, FOXOs, such as FOXO1 and FOXO3, both support and suppress metastatic breast cancer progression[63]. While our data indicate that expression of FOXO4 is regulated at least in part via RdRP activity of hTERT phosphorylated by CDK1, regulation of FOXO4 might be complicated and yet to be elucidated. Further studies are required for a detailed understanding of gene expression regulation by hTERT in mitosis.

## Methods

**Antibodies**. Anti-hTERT mouse monoclonal antibodies (mAbs, clones 10E9-2 and 2E4-2) were generated and described the specificity as reported previously[14].

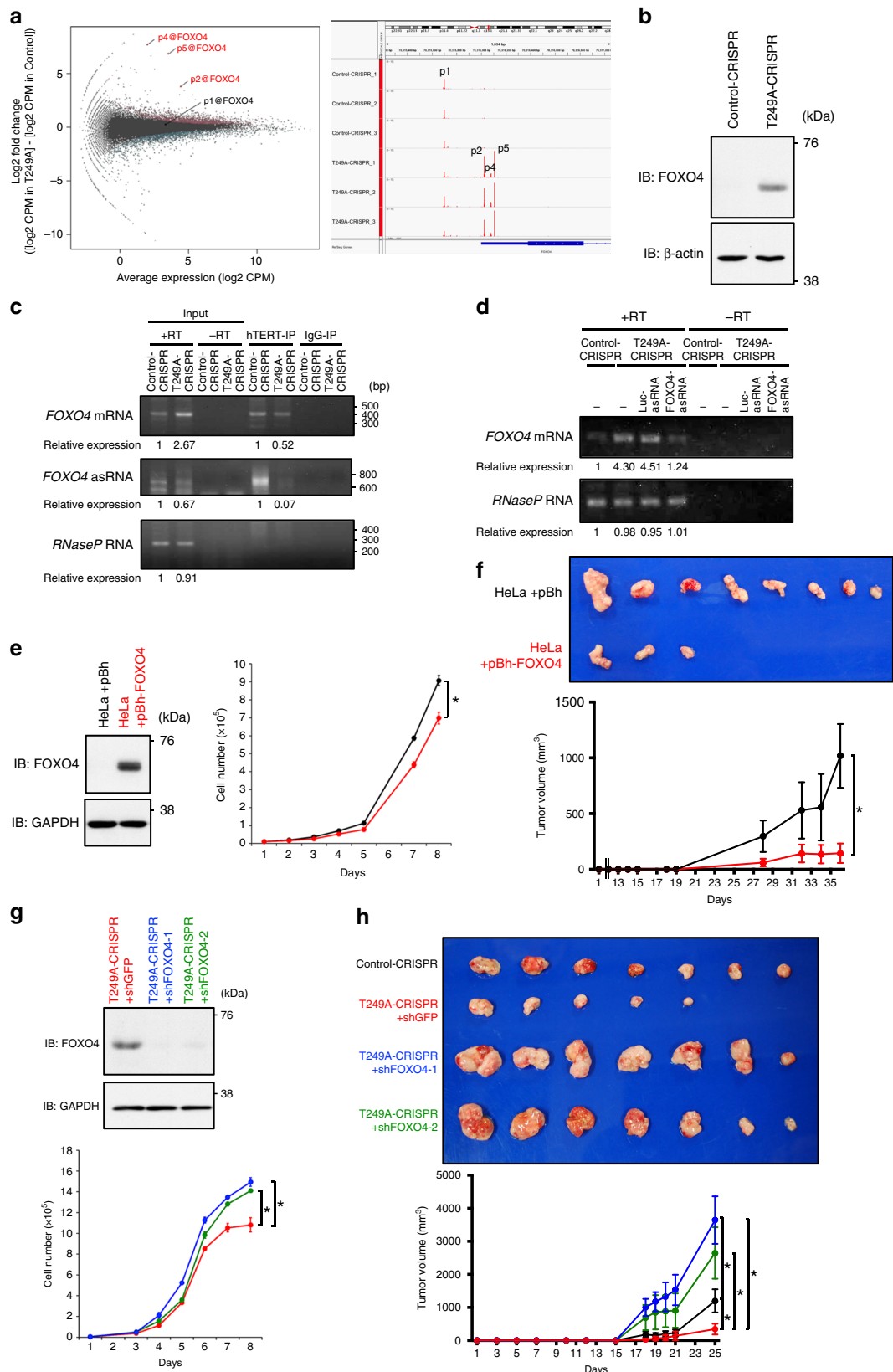

Phosphospecific rabbit polyclonal antibodies (pAbs) and mouse mAb against phosphorylated threonine 249 of hTERT (anti-249T-P and TpMab-1, respectively) were generated in this study (see below). Anti-hTERT mouse mAb (1:100, clone 2E4-2), anti-hTERT rabbit mAb (1:1000, Abcam, ab32020), anti-249T-P pAbs (1:1000), TpMab-1 mAb (1:50), anti-phospho-histone H3 (Ser10) rabbit pAbs (1:2000, Merck, 06-570), anti-cdc2 [CDK1] (POH1) mouse mAb (1:1000, Cell Signaling Technology, 9116), anti-β-actin (AC-15) mouse mAb (1:20000, Sigma-Aldrich, A5441), anti-FOXO4 rabbit mAb (1:4000, Abcam, ab128908), anti-GAPDH mouse mAb (1:1000, MBL Co., Ltd., M171-3), and anti-SMG6 rabbit pAbs[64] (1:1000) were used for immunoblotting (IB). Anti-hTERT mAb (clone 10E9-2: MBL Co., Ltd., M216-3) and anti-hTERT sheep pAbs (Abbexa Ltd., abx120550) were used for immunoprecipitation (IP). Anti-CDK1 rabbit pAbs (1:250, Merck, HPA003387), anti-249T-P pAbs (1:250) and TpMab-1 mAb (1:250) were used for immunohistochemistry.

**Fig. 6 Expression of *FOXO4* is regulated by hTERT phosphorylated on T249. a** MA plot of CAGE peaks differentially expressed between the T249A-CRISPR and Control-CRISPR (left panel). *x*- and *y*-axes indicate average expression levels and fold changes with triplicates, respectively. Individual dots represent CAGE peaks corresponding to promoters defined by the FANTOM5 project[77]. CAGE peaks with statistical significance are indicated by pink and light blue for higher and lower expression in the T249A-CRISPR, respectively. *FOXO4* are indicated with red on MA plot and the positions of each *FOXO4* CAGE peak are visualized by IGV (right panel). **b** The endogenous proteins were detected by anti-FOXO4 mAb in T249A-CRISPR while not in Control-CRISPR. β-actin: an internal control. **c** Immune complexes were isolated with 10E9-2 and associated RNAs were subjected to RT-PCR. **d** Transfection of asRNA complementary to the first exon of *FOXO4* or firefly luciferase mRNA. Cellular RNAs were subjected to RT-PCR. "RT" indicates reverse transcriptase reaction. The relative expressions are noted below the panel (for **c**, **d**). **e** Immunoblotting of ectopically overexpressed FOXO4 (left panel) and cell proliferation (right panel); infected with pBABE-hygro (pBh) (black) and pBh-FOXO4 (red) retroviruses. The cell proliferation assay was done in triplicate and data are shown as the mean ± SD. Asterisk: statistically significant values (*p* < 0.05 by Student's *t*-test, two-sided). **f** Tumor appearances generated from subcutaneous injection in eight mice (upper panel). The volume curves of tumors are demonstrated as the mean ± SEM (*n* = 8) (lower panel). Asterisk: statistically significant values (*p* = 0.0092 by unpaired *t*-test, two-sided). **g** Immunoblotting of FOXO4 (upper panel). Cell proliferation assay (lower panel) was done in triplicate and data are shown as the mean ± SD. Asterisk: statistically significant values (*p* < 0.05 by Student's *t*-test, two-sided). **h** Tumors were established from subcutaneous injection in seven mice (upper panel). Tumor-volume curves of xenografts are shown as the mean ± SEM (*n* = 7) (lower panel). Asterisk: statistically significant values (*p* < 0.05 by unpaired *t*-test). Control-CRISPR (black), T249A-CRISPR+shGFP (red), T249A-CRISPR+shFOXO4-1 (blue), and T249A-CRISPR + shFOXO4-2 (green) (for **g**, **h**). Experiments were repeated three times (for **b**) and twice (for **c**–**e** (left panel), **g** (upper panel)) with similar results. Source data are provided in the Source Data file.

**Generation of phosphospecific polyclonal antibodies**. To generate phosphospecific antibodies against phosphorylated threonine 249 of hTERT (anti-249T-P), hTERT phosphopeptide [244]CEPERpTPVGQG[254] was used to immunize rabbits. The antibodies were purified from the antisera by using hTERT phosphopeptide-conjugated resin and then further purified by passing it through hTERT nonphosphopeptide-conjugated resin.

**Generation of hybridoma producing TpMab-1 antibody**. Female 4-week-old BALB/c mice were purchased from CLEA Japan and kept under specific pathogen-free (SPF) conditions. The Animal Care and Use Committee of Tohoku University approved animal experiments to produce antibody in this study. BALB/c mice were immunized by intraperitoneal (i.p.) injection of 100 μg of hTERT phosphopeptide [244]CEPERpTPVGQG[254] together with Inject Alum (Thermo Fisher Scientific). After two additional immunizations of 100 μg, a booster injection of 100 μg was given i.p. 2 days before spleen cells were harvested. The spleen cells were fused with P3U1 cells (ATCC) using PEG1500 (Roche Diagnostics,). The hybridomas were grown in RPMI 1640 medium (Nacalai Tesque) at 37 °C in a humidified atmosphere containing 5% CO$_2$ and 95% air, supplemented with 10% heat-inactivated fetal bovine serum (Thermo Fisher Scientific), hypoxanthine, aminopterin, and thymidine (HAT) selection medium supplement (Thermo Fisher Scientific), and 5% BriClone Hybridoma Cloning Medium (QED Bioscience). One hundred units/ml penicillin, 100 μg/ml streptomycin, and 25 μg/ml amphotericin B (Nacalai Tesque) were added to the culture medium. Plasmocin (5 μg/ml; InvivoGen) was also used to prevent *Mycoplasma* contamination. The culture supernatants were screened using enzyme-linked immunosorbent assay (ELISA) for binding to hTERT phosphopeptide and nonphosphopeptide.

**Enzyme-linked immunosorbent assay (ELISA)**. Peptides were immobilized on Nunc Maxisorp 96-well immunoplates (Thermo Fisher Scientific) at 1 μg/ml. After blocking with SuperBlock T20 (PBS) Blocking Buffer (Thermo Fisher Scientific), the plates were incubated with culture supernatant with subsequent 1:2000 diluted peroxidase-conjugated anti-mouse IgG (Dako). The enzymatic reaction was conducted with 1-Step Ultra TMB-ELISA (Thermo Fisher Scientific). The optical density was measured at 655 nm using an iMark microplate reader (Bio-Rad Laboratories). These reactions were performed with a volume of 50–100 μl at 37 °C.

**Cell culture and mitotic cell synchronization**. The human cervical carcinoma cell line HeLa, the SV40-transformed human embryonic kidney cell line HEK-293T (293T) and human patient-derived hepatocellular carcinoma cell line MT were cultured in DMEM supplemented with 10% heat-inactivated fetal bovine serum (IFS). The human ovarian carcinoma cell lines PEO1 and PEO14 were cultured in RPMI-1640 medium supplemented with 10% IFS and 2 mM sodium pyruvate (Gibco). The human foreskin fibroblast cell line BJ was cultured in K/O DMEM/Medium 199 (4:1) supplemented with 15% IFS and L-Gultamine (Nacalai Tesque).

Mitotic cell synchronization was performed by the following method. Cells were switched to medium containing 2.5 mM thymidine (Nacalai Tesque) and incubated for 24 h. Six hours after release, cells were incubated in medium containing 0.1 μg/ml nocodazole (Sigma-Aldrich) for 16 h. After shake-off, mitotic cells were retrieved[14]. Cells arrested in mitosis with nocodazole were confirmed by IB using anti-phospho-histone H3 (Ser10) antibodies.

**Transfection of siRNAs**. For suppression of hTERT or CDK1 expression, HeLa cells were transfected with siRNAs using Lipofectamine 2000 (Thermo Fisher Scientific). After 48 h of incubation, cells were treated with 0.1 μg/ml nocodazole

for 16 h. The sequences of siRNAs against hTERT[16] are listed in Supplementary Table 7. MISSION siRNAs Hs_CDC2_4049_s and Hs_CDC2_4049_as, Hs_CDC2_4053_s and Hs_CDC2_4053_as, Hs_CDC2_6210_s and Hs_CDC2_6210_as, and Hs_CDC2_4050_s and Hs_CDC2_4050_as were used for CDK1 siRNA #1, CDK1 siRNA #2, CDK1 siRNA #3 and CDK1 siRNA #4, respectively. MISSION siRNA Universal Negative Control #1 (Sigma-Aldrich) was used as a negative control.

**Statistics and reproducibility**. Error bars in graphs represent SD or SEM as indicated in the figure legends. The "*n*" numbers in the legends indicate biologically independent samples (cells or animals) and statistics such as error bars are derived from only the case of *n* ≥ 3. Statistically significant differences between conditions were determined using two-sided unpaired *t*-tests or as specified in figure legends. Statistical significance is represented as *p*-values.

**IP-IB of hTERT**. In all, $1 \times 10^7$ cells were lysed in 1 ml of Lysis buffer A (0.5% NP-40, 20 mM Tris-HCl (pH 7.4), and 150 mM NaCl). After sonication, lysates were cleared of insoluble material by centrifugation at $21{,}000 \times g$ at 4 °C for 15 min. One milliliter of lysate was pre-absorbed with 40 μl of Pierce Protein A Plus Agarose (Thermo Fisher Scientific) for 30 min at 4 °C. Pre-absorbed lysate was mixed with 10 μg of anti-hTERT mAb (clone 10E9-2)[14] or 30 μg of anti-hTERT pAbs (abx120550) and 40 μl of Pierce Protein A Plus Agarose, and incubated overnight at 4 °C. Immune complexes were washed three times with Lysis buffer A and eluted in 2× SDS loading buffer (2% β-mercaptoethanol, 20% glycerol, 4% SDS, and 100 mM Tris-HCl (pH 6.8)), and then subjected to SDS-PAGE in 8% polyacrylamide gels. Anti-hTERT mouse mAb (clone 2E4-2) and Mouse TrueBlot ULTRA Anti-Mouse Ig HRP (Rockland) or anti-hTERT rabbit mAb (ab32020) and anti-rabbit IgG HRP (GE Healthcare) were used for IB to detect whole-hTERT proteins. Anti-phospho hTERT rabbit pAbs (anti-249T-P) and anti-rabbit IgG HRP (GE Healthcare) or anti-phospho hTERT mouse mAb (TpMab-1) and anti-mouse IgG HRP (GE Healthcare) were used for IB to detect phosphorylated hTERT proteins.

For the λ phosphatase treatment, the beads suspension with immune complexes was treated with 2000 U of λ protein phosphatase (Bio Academia) and 2 mM MnCl$_2$ in λ-PPase reaction buffer (50 mM Tris-HCl (pH 7.6), 100 mM NaCl, 2 mM DTT, 100 μM EDTA and 0.01% Brij 35) and incubated at 30 °C for 30 min.

**Identification of phosphopeptides by MS analysis**. The protein samples immunoprecipitated with anti-hTERT mAb (clone 10E9-2) were separated by 12.5% SDS-PAGE and subjected to in-gel digestion using trypsin[65]. The tryptic digests were subjected to liquid chromatography coupled with nanoelectrospray tandem mass spectrometry (Finnigan LTQ Orbitrap XL mass spectrometer; Thermo Fisher Scientific). The Mascot software package (version 2.5.1; Matrix Science) was used to search for the mass of each peptide ion peak against the SWISS-PROT database (Homo sapiens, 20,205 sequences in the Swiss prot_2015_09.fasta file) using the following parameters: 1 missed cleavage; fixed modification: carboxymethylation (C); variable modification: oxidation (M), phosphorylation (ST), phosphorylation (Y); search mode: MS/MS ion search with decoy database search included; peptide mass tolerance ± 10 ppm; MS/MS mass tolerance ± 0.8 Da; peptide charge: 2+ and 3 + .

**In vitro kinase assay using recombinant hTERT proteins**. For recombinant protein expression using cell-free synthesis-coupled transcription–translation[66], the target cDNA fragment, corresponding to 191–306 residues of hTERT was subcloned into the pCR2.1-TOPO vector (Thermo Fisher Scientific). Proteins, subjected to affinity purification, were N-terminally fused with a modified natural

polyhistidine N11-tag (amino-acid sequence: MKDHLIHNHHKHEHAHAEH) with a TEV (Tobacco Etch virus) protease recognition site and a GSSGSSG linker sequence. These sequences were introduced using TOPO cloning. The cell-free synthesized N11-tagged hTERT_191-306 protein was purified using an AKTA 10 S system (GE Healthcare) with a HisTrap columns (GE Healthcare); the AKTA 10 S system was washed with a concentration gradient buffer (50 mM Tris-HCl buffer at pH 8.0, containing 1 M NaCl and 10 mM imidazole). The N11-tagged recombinant proteins were eluted with a concentration gradient of imidazole (from 10 to 500 mM) in elution buffer (50 mM Tris-HCl buffer at pH 8.0, containing 0.5 M NaCl). Imidazole was removed by overnight dialysis at 4 °C in wash buffer. The affinity-tags were removed by incubation at 4 °C for 20 h with TEV protease. The resulting tag-cleaved proteins were purified by ion exchange chromatography with a HiTrap SP column (GE Healthcare) and gel filtration with the final buffer (25 mM Tris-HCl buffer at pH 7.0, containing 450 mM NaCl, 0.25 mM TCEP) using HiLoad 16/ 600 Superdex columns (GE Healthcare).

For baculovirus–insect cell expression of active form of human IKK2 (IKK2_2-664), we used the Bac-to-Bac Baculovirus Expression System (Thermo Fisher Scientific). The target cDNA fragment, corresponding to 2-664 residues of human IKK2, was sub-cloned into the pDEST vector (Thermo Fisher Scientific). Polyhistidine affinity-tagged IKK2_2-664 was expressed in insect Sf9 cells at a multiplicity of infection (MOI) of 1.0 with a recombinant baculovirus that expresses IKK2_2-664. The cells from each culture were harvested 48 h post-infection, and the cell pellets were washed once with phosphate-buffered saline (PBS) and were immediately frozen in liquid N2. The recombinant IKK2_2-664 protein was purified to use as a negative control in vitro kinase assay[67].

Purified hTERT_191-306 proteins were incubated with CDK1-cyclinB (New England Biolabs) or purified IKK2_2-664 proteins. In vitro kinase assays were carried out in 30 μl of kinase reaction buffer (50 mM Tris-HCl pH7.5, 10 mM MgCl₂, 0.1 mM EDTA, 2 mM DTT, 0.01% Brij 35) containing 4 mM ATP, 10 μM of hTERT_191-306 proteins, and 30 units of CDK1-cyclinB or 1 μM of IKK2_2-664. Reactions were incubated at 37 °C for 2 h. Reaction samples were terminated by adding 2.5x SDS sample loading buffer (5% (w/v) SDS, 250 mM DTT, 15% (v/v) glycerol, 140 mM Tris-HCl pH 6.8, and 0.01% (w/v) bromophenol blue), boiled and subjected to Phos-tag SDS-PAGE[22]. There are no specific molecular weight markers that is available for this assay.

For mass spectrometry analyses, gel regions, containing proteins, were excised and digested with trypsin (Promega) for 20 h at 37 °C. The resulting peptides were analyzed by LC-ESI-MS/MS (liquid chromatography-electrospray ionization tandem mass spectrometry) at the Support Unit for Bio-Material Analysis in RIKEN CBS Research Resources Center.

**In vitro kinase assay using endogenous hTERT proteins.** TERT protein was immunoprecipitated from human cell lines as described for the IP-IB assay with anti-hTERT mAb (clone 10E9-2). The beads suspension with immune complexes was treated with 2000 U of λ protein phosphatase (Bio Academia) and 2 mM MnCl₂ in λ-PPase reaction buffer (50 mM Tris-HCl (pH 7.6), 100 mM NaCl, 2 mM DTT, 100 μM EDTA and 0.01% Brij 35), and incubated at 30 °C for 30 min. Immune complexes were washed three times with Lysis buffer A and twice with kinase reaction buffer (50 mM Tris-HCl pH7.5, 10 mM MgCl₂, 0.1 mM EDTA, 2 mM DTT, 0.01% Brij 35) containing 1x PhosStop phosphatase inhibitors (Sigma). In vitro kinase assays were carried out in 20 μl of kinase reaction buffer containing 1x PhosStop, 6.5 mM ATP, and 0.2 μg of CDK1-cyclinB (Carna Biosciences). Reactions were incubated at 37 °C for 2 h. Reaction samples were washed once with kinase reaction buffer containing 1x PhosStop, added 2x SDS sample loading buffer, boiled and subjected to SDS-PAGE in 8% polyacrylamide gels.

**IP-RdRP assay.** TERT protein was immunoprecipitated from human cell lines as described for the IP-IB assay with anti-hTERT mAb (clone 10E9-2)[14]. The bead suspension with immune complexes was washed four times with 1× acetate buffer (10 mM HEPES-KOH (pH 7.8), 100 mM potassium acetate, and 4 mM MgCl₂) containing 10% glycerol, 0.1% Triton-X, and 0.06× cOmplete EDTA-free (Roche), and once with AGC solution (1× acetate buffer containing 10% glycerol and 0.02% CHAPS) containing 2 mM CaCl₂. The bead suspension was treated with 0.25 unit/ μl Micrococcal Nuclease (Takara Bio) at 25 °C for 15 min. Immunoprecipitates were subsequently washed twice with AGC solution containing 3 mM EGTA and once with 1× acetate buffer containing 0.02% CHAPS. Forty microliter of reaction mixture was prepared by combining 20 μl of the bead suspension with 6 μl of [α-³²P] UTP (3,000 Ci/mmol) and 25 ng/μl (final concentration) of RNA template, and incubated at 32 °C for 2 h. The sequence of RNA template is as follows: 5′-GG GAUCAUGUGGGUCCUAUUACAUUUUAAACCCA-3′[68]. This RNA has hydroxyl groups at both the 5′ and 3′ ends. The final concentrations of ribonucleotides were 1 mM ATP, 0.2 mM GTP, 10.5 μM UTP, and 0.2 mM CTP. The resulting products were treated with Proteinase K to stop the reaction, purified several times with phenol/chloroform until the white interface disappeared, and precipitated using ethanol. The RdRP products were treated with RNase I (2 U, Promega) at 37 °C for 2 h to digest single-stranded RNAs completely, followed by Proteinase K treatment, phenol/chloroform purification, and ethanol precipitation. The products were electrophoresed in a 10% polyacrylamide gel containing 7 M urea, and detected by autoradiography. The signal intensities were quantified using Fiji/ImageJ software.

**Generation of phosphorylation mutant plasmids.** For site-directed mutagenesis, QuikChange II XL site-directed mutagenesis Kit (Agilent Technologies) was used. In brief, site-directed mutagenesis PCRs were performed using PfuUltra High-Fidelity DNA polymerase following the manufacturer's protocol with either the pBABE-puro-hTERT retroviral vector or the pNK-FLAG-Z-hTERT expression vector as template plasmids. Sequences of mutagenic primers are listed in Supplementary Table 7. PCR products were digested with DpnI restriction enzyme for 1 h at 37 °C and then transformed into XL-10-Gold ultracompetent cells. Mutations were confirmed by Sanger sequencing.

**Telomere analysis.** For the telomeric repeat amplification protocol (TRAP) assay, $1 \times 10^5$ cells were suspended in 200 μl of TRAP lysis buffer (10 mM Tris-HCl (pH 7.5), 1 mM MgCl₂, 1 mM EGTA, 0.5% CHAPS, 10% glycerol, 100 μM Pefabloc SC, and 0.035% 2-mercaptoethanol). The TRAP assay was performed with 5 μl of the suspension by a conventional method[16].

To measure telomere length by Southern blotting, genomic DNAs were isolated, digested with HinfI and AfaI, electrophoresed in a 0.8% agarose gel and hybridized with a ³²P-labeled telomeric (CCCTAA)₃[69].

**Direct telomerase assay.** For the direct telomerase assay, we modified the original methods[17,18]. Briefly, TERT protein was immunoprecipitated from human cell lines as described for the IP-IB assay without sonication. Immune complexes were washed three times with Lysis buffer A, and then suspended in 30 μl of TRAP lysis buffer. The direct telomerase assay was carried out with 10 μl of the suspension and 40 μl of reaction mixture (2.5 μl of [α-³²P] dGTP (6000 Ci/mmol), 50 mM Tris-HCl (pH 8.0), 50 mM KCl, 1 mM MgCl₂, 1.25 mM Spermidine, 5 mM 2-mercaptoethanol, 2.5 mM dTTP, 2.5 mM dATP, 25 μM dGTP and 2.5 μM a5 primer (5′-TTAGGGTTAGGGTTAGCGTTA-3′)) by incubation at 37 °C for 2 h. 5′-³²P-labeled synthetic 18-mer DNA was added as an internal recovery/loading standard[17]. The products were purified with phenol/chloroform, and precipitated using ethanol. The products were electrophoresed in a 10% polyacrylamide gel containing 7M urea, and detected by autoradiography. The signal intensities were quantified using Fiji/ImageJ software.

**Generation of stable cell lines and proliferation assay.** Amphotropic retro-viruses were created using the retroviral vectors pBABE-puro, pBABE-puro-hTERT T249A or pBABE-puro-hTERT T249E for making stable cell lines expressing hTERT mutants. After infection, polyclonal cell populations were purified by selection with puromycin (2 μg/ml) for 3 days[70].

We used the following short-hairpin RNA (shRNA) vectors, shFOXO4-1 (TRCN0000010291) and shFOXO4-2 (TRCN0000039720), constructed by The RNAi Consortium and the sequences are listed in Supplementary Table 7. shGFP was used as the control. These vectors were used to make amphotropic lentiviruses, and the cell populations were selected with puromycin (2 μg/ml) for 3 days.

For generating stable cell lines expressing FOXO4, we constructed the retroviral vector pBABE-hygro (pBh)-FOXO4. After infection of pBABE-hygro or pBh-FOXO4, the cells were selected with hygromycin B (200 μg/ml) for 7 days.

To generate proliferation curves, cells were plated in triplicate and counted in a Z2 Particle Count and Size Analyzer (Beckman-Coulter).

**RT-PCR and quantitative reverse transcription PCR (qRT-PCR).** Total cellular RNA was isolated using TRIzol (Thermo Fisher Scientific), treated with RQ1 DNase (Promega), and subjected to RT-PCR and qPCR. For IP-RT-PCR, RNA samples were extracted from the immune complexes as described for the IP-IB assay without sonication. Immune complexes were washed three times with Lysis buffer A containing 300 mM NaCl and associated RNAs were isolated with TRIzol. The RT reaction was performed with oligo(dT)₁₂₋₁₈ (Thermo Fisher Scientific) primer or target- and strand-specific primers using PrimeScript Reverse Transcriptase (Takara Bio) for 60 min at 42 °C, followed immediately by PCR. qPCR was performed with a LightCycler 480 II (Roche) using LightCycler 480 SYBR Green I Master (Roche) according to the manufacture's protocols. Sequences of PCR primers are listed in Supplementary Table 7.

**Genome editing using CRISPR-Cas9.** The PuroCas9 plasmid, which consists of three cassettes for Cas9 cDNA, chimeric guide RNA, and puromycin cDNA, was generated by modifying the px330 plasmid (addgene #42230)[71]. Cas9 and pur-omycin resistance genes are expressed under the control of the CAG promoter[72] and the PGK promoter[73], respectively. The guide RNA sequence against hTERT was cloned into the BbsI site of the PuroCas9 plasmid (PuroCas9-hTERT). For generating T249A-CRISPR and T249E-CRISPR, the T249A donor plasmid or T249E single-stranded oligo included the point mutation, leading to amino-acid change were transfected with PuroCas9-hTERT using FuGeneHD (Promega) into 293T cells, respectively. For generating Control-CRISPR, PuroCas9 plasmid was transfected using FuGeneHD into 293T cells. Following selection with puromycin, single-cell was cloned. Genomic DNA was extracted with the GenElute Mamma-lian genomic DNA miniprep kit (Sigma) according to the manufacturer's instructions. Presence of mutation in single-cell clones was confirmed by PCR amplification, restriction enzyme digestion and Sanger sequencing. Potential off-target (OT) mutation sites of the guide RNA were predicted[74] and verified the

sequences by Sanger sequencing following PCR amplification using OT site-specific primers. Sequences of PCR primers, donor plasmid/oligo and guide RNA are listed in Supplementary Table 7.

**Xenotransplantation.** Six-week-old male Non-obese diabetic/severe combined immunodeficiency (NOD/SCID) mice (NOD/NCrCRL-Prkdc$^{scid}$) were purchased from Charles River Laboratories, Inc. and used as recipients for xenotransplantation. They were maintained in SPF rooms at 20 °C, 50% humidity, and 12-h light/12-h dark cycle condition. $1 \times 10^5$ cells were suspended in a mixture of serum-free medium and Matrigel (BD Biosciences; 1:1 volume). The mixture was injected subcutaneously through a 26-gauge needle into the right dorsal areas of anesthetized NOD/SCID mice. We monitored tumor formation and tumor size every two or three days, and dissected out the tumors within a month after engraftment.

**CAGE sequencing and data analysis.** Of each of the total RNAs extracted from two triplicates (Control-CRISPR and T249A-CRISPR), 3 µg was used to prepare a sequencing library according to the non-Amplified non-Tagging Illumina Cap Analysis of Gene Expression (nAnT-iCAGE)[29] by using CAGE library preparation kit (DNAFORM), and sequenced by NextSeq500 platform (Illumina). After discarding sequences with ambiguous (N) and low-quality (Phred < 30), the remaining reads were aligned with the human reference genome (GRCh37) by STAR v2.6.0a[75] with a guide of known junctions of RefSeq transcripts[76]. The alignments with mapping quality more than 20 were selected, and their 5′-ends were counted based on the robust set of CAGE peaks identified in a previous study[77] and provided at the FANTOM5 web resource[78]. The read counts were normalized as counts per million (CPM) with relative log expression (RLE) method[79]. The sequencing results were visualized using Integrative Genomics Viewer (IGV)[80]. This normalization and subsequent differential analysis were conducted with edgeR[81], where FDR < 1e-4 were used as a criteria for statistical significance. $p$-value of GO term enrichment was assessed with DAVID v6.8[82] using all human genes as a background set, and only the terms that have less than 100 corresponding genes were chosen to exclude very general GO terms such as "biological process".

**Clinical pancreatic tissue samples.** The pancreatic tissues used in this study were obtained from patients who underwent surgical treatment at Tokyo Metropolitan Geriatric Hospital ($n = 47$; female, $n = 25$; male, $n = 22$; 62–91 years old; mean age, $74.8 \pm 7.0$ years old) and all patients provided written informed consent. The present study was conducted in accordance with the principles embodied in the Declaration of Helsinki, 2013, and all experiments were approved by the ethics committees of Tokyo Metropolitan Geriatric Hospital and Institute of Gerontology (permit-#260219).

Tissues were fixed in 10% buffered formalin and then subjected to standard tissue processing and paraffin embedding. The tissues were sliced serially into sections 3 µm thick for hematoxylin and eosin (H&E) and immunohistochemical staining. Pathological specimens were diagnosed by our pathologists based on the World Health Organization Classification of Tumors of the Digestive System[83]. The most common precursor lesions of pancreatic ductal adenocarcinoma are PanIN lesions. PanINs are microscopic papillary or flat, noninvasive epithelial neoplasms that are usually <5 mm in diameter and confined to pancreatic ducts. PanINs are divided into three grades according to the degree of cytological and architectural atypia. Lesions with minimal, moderate or marked atypia are designated PanIN-1, PanIN-2, and PanIN-3, respectively. PanIN-1A is flat epithelial lesion composed of tall columnar cells with basally located nuclei and abundant supranuclear mucin. PanIN-1B has a papillary, micropapillary, or basally pseudostratified architecture but are otherwise identical to PanIN-1A. PanIN-2 (pancreatic intraepithelial neoplasia 2) may be flat but are mostly papillary. Cytologically, by definition, PanIN-2 must have some nuclear abnormalities. PanIN-3 is usually papillary or micropapillary; however, they may rarely be flat. True cribriforming, the appearance of "budding off" of small clusters of epithelial cells into the lumen, and luminal necrosis should all suggest the diagnosis of PanIN-3. PanIN-3 resembles carcinoma at the cytonuclear level, but invasion through the basement membrane is absent[84].

**Immunohistochemistry (IHC) of pancreatic tissue samples.** Paraffin-embedded sections (3 µm) were subjected to immunostaining. After deparaffinization, the tissue sections were preheated in HEAT PROCESSOR Solution pH 6 (Nichirei) for 20 min at 100 °C. Then, sections were incubated for 5 min at room temperature with Protein Block Serum-Free (Dako). The tissue sections were then incubated with the anti-249T-P pAbs, TpMab-1 mAb (1: 250 in dilution) or anti-CDK1 pAbs (1: 250 in dilution) for 1 h at room temperature. Endogenous peroxidase activity was blocked by 3% $H_2O_2$ for 5 min at room temperature. Sections were incubated with Second antibody (REAL EnVision, Dako). Bound antibodies were detected using diaminobenzidine tetrahydrochloride as the substrate. The sections were then counterstained with Mayer's hematoxylin. Negative control tissue sections were prepared by omitting the primary antibody. As for the evaluation of immunostaining, proportion of positively stained nucleus of cancer cells were analyzed at magnification x200. For statistical analysis, the patient groups with 249T-P and CDK1 positive/negative were divided at 10% cutoff value.

**Fluorescent staining of pancreatic tissue samples.** Paraffin-embedded sections (3 µm) were subjected to fluorescent staining. Fluorescent labeling of primary antibodies were performed using conjugation kit (abcam). After deparaffinization, the tissue sections were incubated overnight at 4 °C with anti-CDK1 pAbs (1:100 in dilution) and TpMab-1 mAb (1:100 in dilution). After incubation, the sections were mounted with Vectashield H-1200 containing DAPI. Fluorescent images were captured by a CCD camera (ORCA-ER-1394, Hamamatsu Photonics KK) mounted on a microscope (80i, Nikon). Intensity of fluorescence was analyzed using Win-ROOF2015 (Mitani Corporation).

**Statistical analysis of pancreatic tissue sample.** The level of significance was set at $p < 0.05$ for all analyses. Statistical analyses were performed using the StatView J version 5.0 software package (SAS Institute) and SPSS version 22 (IBM Corp.).

**Clinical HCC samples.** A total of 100 HCC patients who received surgery at Kanazawa University Hospital from 2008 to 2013 were enrolled in the study. This study was approved by the Institutional Review Board at Kanazawa University (IRB # 1065) and all patients provided written informed consent.

**IHC of HCC samples.** IHC of HCC tissues and adjacent non-cancerous liver tissues was performed using DAKO Envision+ kits (Agilent Technologies) according to the manufacturer's instruction. Briefly, formalin-fixed paraffin-embedded tissue slides were deparaffinized, rehydrated, and immediately proceeded for antigen retrieval (120 °C for 5 min) using autoclaves and target retrieval solution, citrate pH 6 (Agilent Technologies). Slides were immersed with blocking solution (Agilent Technologies) for 15 min and subsequently replaced with anti-249T-P pAbs or anti-CDK1 pAbs diluted at 1:250 with antibody diluent solution (Agilent Technologies). Slides were incubated at 4 °C overnight and then washed and visualized with DAB + substrate chromogen (Agilent Technologies). The patient groups with 249T-P and CDK1 positive/negative were divided at 10% cutoff value.

**Statistical analysis of HCC samples.** Kaplan–Meier survival analysis was performed in GraphPad Prism software 6.0 (GraphPad Software). The association of phosphorylation of hTERT threonine 249 and clinicopathologic characteristics was examined with either Student's tests or $\chi^2$ tests.

**Transfection of *FOXO4*-asRNA.** An asRNA complementary to the first exon of *FOXO4* mRNA was synthesized using T7 RNA polymerase. A part of the first exon of *FOXO4* mRNA was amplified by PCR and the PCR amplicon was used as the template for in vitro transcription by T7 RNA polymerase (Promega). After the in vitro transcription reaction, the transcribed RNA was treated by DNase (RQ1 DNase; Promega) and purified with MicroSpin G-25 column (GE Healthcare) and phenol/chloroform. An asRNA complementary to firefly luciferase mRNA was similarly synthesized using a firefly luciferase-expressing plasmid (pGL4.10; Promega). The sequences of the primers used are shown in Supplementary Table 7. The synthesized asRNA were transfected in T249A-CRISPR cells using Lipofectamine 2000 (Thermo Fisher Scientific).

**Reporting summary.** A reporting summary for this article is available as a Supplementary Information file

## Data availability

The CAGE sequencing data reported in this study are available in the DDBJ Sequence Read Archive (DRA) with accession number (DRA007587). All peptide information of TERT proteins detected by mass spectrometry analysis in this study is shown in Supplementary Data 1 and Supplementary Table 1. The source data underlying Figs. 1a, b, c, f, g, h, i, 3a, b, c, d, e, f, 4a, b, c, d, 5b, d, f, 6b, c, d, e, g, and Supplementary Figs. 1a, b, c, d, 2a, b, c, 8a, b, c, 12a, b, c, d, 13, 14, 15, 16, 18a, 19a 20a, 20b are provided as the Source Data file. All the other data supporting the findings of this study are available within the article and its supplementary information files and from the corresponding author upon reasonable request. A reporting summary for this article is available as a Supplementary Information file.

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

## Acknowledgements
We would like to thank Yasuhiro Murakawa for useful discussions and Saori Ueki for technical supports. We are grateful to the Support Unit for Bio-Material Analysis, RIKEN CBS RRD, for technical help with the mass spectrometry analysis. This work was supported in part by the Grant-in-Aid for challenging Exploratory Research under Grant Number 15K14482, AMED under Grant Number JP18fk0210005, The National Cancer Center Research and Development Fund (30-A-4), Daiichi Sankyo Foundation of Life Science to K.M. and the Platform Project for Supporting Drug Discovery and Life Science Research (Basis for Supporting Innovative Drug Discovery and Life Science Research [BINDS]) from AMED under Grant Number JP17am0101078 to Y.K.

## Author contributions
M.Y., Y.A., and M.M. performed biochemical and cellular experiments. T.Y., Y.M., and S.K. performed immunohistological and statistical analysis of clinical samples. S.S. and M.S. designed and performed in vitro kinase assay. M.S.M., H.K., and Y.H. performed CAGE sequencing and the bioinformatics analyses. K.S. and T.K. carried out mass spectrometric protein profiling and the bioinformatics analyses. T.A. and Y.F. designed the experiments for CRISPR-Cas9 gene editing of hTERT. S.Y., M.K.K., and Y.K. generated a phosphospecific monoclonal antibody against hTERT. M.Y., Y.A., T.Y., and K.M. designed the experiments, discussed the interpretation of the results and wrote the manuscript. All authors read and approved the final manuscript.

## Competing interests
The authors declare no competing interests.
