## [Peer Review File · Nature Communications]

Reviewers' comments:

Reviewer #1 (Remarks to the Author): Expertise in CDK1

Manuscript "CDK1 dependent phosphorylation of hTERT contributes to cancer progression" by Yasukawa, Masutomi and colleagues claims that the mitotic kinase CDK1 is the key regulator of telomerase reverse transcriptase hTERT RNA dependent RNA polymerase (RdRP) activity, but not of its telomerase activity. Introducing a phospho-mimetic mutation (T249E) into endogenous hTERT inhibited the tumorigenic potential of human cancer cell lines. The authors also show that phosphorylation of hTERT T249 negatively regulates transcription factor forkhead box O4 (FOXO4) expression, and propose a model involving degradation of double-stranded RNAs leading to cancer progression.

The study is technically sound and contains nice multiple-level proofs for the proposed model. However, there are few important points outlined below that should be addressed before publishing. These mostly concern the direct correlation of T249 phosphorylation with elevated CDK1 activity, and validation of glutamic acid as a true phospho-mimetic in this particular context.

- 1) The authors should provide direct evidence of CDK1 phosphorylation effect on hTERT RdRP activity. While they showed that expression of the phosphorylation-defective mutant hTERT T249A decreased RdRP activity, and hTERT T249E mutant increased RdRP activity, suggesting that this mutant acts as a phosphomimetic mutant, they did not provide direct evidence that CDK1 phosphorylated T249 does the same in vitro. It would be easy to use the phosphatase to remove the phosphate groups from hTERT proteins, like in Figure 3A, and then get rid of the phosphatase and add CDK1 and ATP to restore the phosphorylation. And then, perform the assays again. This would support the claim that the T249E really acts as a true mimetic for CDK1 phosphorylation, and would allow to postulate with sufficient certainty, that phosphorylation of hTERT T249 is required for the RdRP activity but not for reverse transcriptase activity of hTERT.
- 2) The authors should provide proof that the used cancer models with higher RdRP activity, and individual cells immuno-stained for phospho-T249, really had higher CDK1 activity. They only state that previous reports have demonstrated that CDK1 expression is upregulated in many human malignant tumor tissues and that CDK1 activity correlates with the prognosis of patients with tumors. However, the study does not provide any direct experimental demonstration for such a correlation between CDK1 activity and phosphorylated T249 in tumor cells. They also claim that "the phosphorylation occurs more frequently in aggressive and advanced cancers in clinical samples". Is this higher frequency correlated with higher CDK1 activity relatively to less advanced cancers?
- 3) The following statement on page 8 needs more explanation: "We noted that the sequence surrounding T249 is part of a motif that is frequently phosphorylated in mitosis and has been implicated as a target of the serine-threonine kinase cyclin-dependent kinase 1 (CDK1)." What does "frequently phosphorylated in mitosis" exactly mean?
- 4) On Figure 3B, a control blot for hTERT protein levels should be presented for each lane.
- 5) It is not correct to claim on page 17, first paragraph, that, taken together, these observations confirmed that phosphothreonine 249 in hTERT is essential for cancer cell proliferation through activation of RdRP activity of hTERT. Although in Figure 5E, the effect on tumor volume in this particular xenograft model was quite impressive, indeed, the other experiments in the study do not allow such absolute generalization of essentiality of this particular phosphorylation event for cancer cell proliferation.
- 6) Please clarify how these two statements fit together: (1) Page 24 second paragraph. "Several laboratories have shown that hTERT is phosphorylated. Specifically, four phosphorylation sites have been reported." (2) Later, on the same page. "These phosphorylation sites were identified from consensus sequences for specific kinases, and MS analysis was not conducted in previous reports." Does this mean that, in fact, the previous works did not demonstrate the phosphorylation of hTERT, but rather just predicted based on consensus sequences? There must be more to it, but as it is, the whole paragraph leaves the reader into total confusion. Moreover, the authors themselves state that they were unable to detect these sites by MS analysis in this study with

mitotic cells. Further on, the authors speculate that the amount and level of phosphorylation in these sites may be difficult to detect using MS analysis. This leaves an impression that we can neglect these reports on phosphorylation, as if there is no quantitative MS phosphoproteomics data, it is hard to prove that the predicted phosphorylation events matter. The authors should elaborate on the experiments in literature more to give a clear picture of the evidence for a reader.

7) In connection with the previous point: Phosphate Affinity SDS PAGE western blotting of the IP from nocodazole arrested cells (samples w/o phosphatase like in Figure 1c, for example) would help to see if there are several phosphorylation events, or just one, occurring in vivo.

Minor points

1) Mentioning of serine 227 on page 7 and Figure 1 is out of context and confusing, as it was not detected in MS in Figure 1d. Later mentioning in the main text with the literature reference is sufficient.

2) Page 25/26: "When we introduced substitutions of T249 (T249A and T249E) into endogenous hTERT, we found that these mutants did not affect telomerase activity at telomeres but instead affected RdRP activity and inhibited the tumorigenic potential of human cancer cell lines." The inhibition claim should be only for T249A, not for T249E?

Reviewer #2 (Remarks to the Author): Expertise in telomerase

The group of Masutomi has published on the non-telomere role of hTERT having RdRP activity in cancer cells. In this report they discovery and extensively characterize a phosphosite at T249 of hTERT. This residue is encoded by exon 2 of hTERT which contains the key residues responsible for RNA binding activity of TERC. They confirm previous findings from their group that RdRP activity of TERT is most abundant during mitosis using nocodazole treatment. They extensively characterize key reagents and determine that these reagents are specific for TERT detection, which is good information for the field (telomere/telomerase field). They find that CDK1 a key kinase that phosphorylates hTERT at T249. They show this using multiple strategies. Further, they utilize multiple cancer types and clinically relevant samples showing the importance of this phosphorylation event in clinically advanced cancers. Overall the data are sound and interpreted carefully. That being said I have a few suggestions as outlined below that could improve the impact of the manuscript.

Does the CDK1 inhibitor (RO-3306) impact telomerase activity by TRAP assay or direct telomerase assay?

Can the authors provide evidence in the BJ cells that overexpressed WT TERT has RdRP activity? Using the T249A and T249E cell lines already generated and performing the RdRP assay should be sufficient. I ask because I am curious if normal cells have the needed cofactors for this alternative function of hTERT or if this is something that is utilized solely by cancer cells. This is critical information. It would appear from the clinical samples that the authors think that the answer would be no, normal cells with OE WT TERT do not have RdRP activity because T249 is not readily phosphorylated and they show that this even is critical for RdRP activity in the manuscript. Thus this experiment is critical to provide evidence to the authors idea that advanced cancers more readily utilize this alternative function of TERT.

Why did the authors choose CAGE seq to analyze expression differences in the CRISPR lines? By my understanding of CAGE seq this is used to map TSS of long-lived mRNAs. This could significantly impact the results. While FOXO4 seems to be a strong candidate, could the authors provide a better rationale as to why CAGE seq was chosen over other RNA seq methods?

The hTERT RNA IP assays could be improved by showing a direct interaction of hTERT with

FOXO4A exon 1. If this interaction is direct the authors should be able to crosslink using UV and then IP with hTERT and measure Exon 1 FOXO4 RNAs. Have the authors tried this experiment? The washing in a non-crosslinked IP is not sufficient to rule out cell mixing issues during lysis.

Could the authors treat WT TERT cancer cells with the suspected anti-sense FOXO4 exon 1 RNA that would be generated from the TERT binding and producing RdRP antisense transcripts? This experiment would provide further evidence that this is indeed the mechanism of action. The overexpression and knockdown experiments are a nice start but the piece of data that is missing is the treatment of cells with FOXO4 antisense E1 transcripts that should also knockdown FOXO4 in the fashion the authors are proposing.

Given that the 249T amino acid is not in the TERC RNA binding domain it makes sense that this residue is dispensable for TERC binding but did the authors confirm this in their CRISPR assays? I know the direct assay still worked and TRAP but this would confirm those findings and the authors should already have the IP cDNAs to do the assay. Using the IP samples the authors should confirm hTERT binding to mutant TERT.

Second question along this line of thought is from the 2009 Nature paper by this same group ALU RNAs were the second most abundant RNA found to IP with TERT. Does the mutant still bind ALUs? ALU transcripts are associated with gene expression regulation as well and could further explain the growth phenotypes observed.

Do the authors speculate that in cells that appear to be addicted to TERT, that is die rapidly upon TERT inhibition by siRNAs, that activity is responsible for those findings? It would appear to be the case.

CDK1 is upregulated in many cancer types. Pan-CDK inhibitors are in clinical trials. Can the author provide insights into using CDK1 inhibition in telomerase positive cancers and if there are better inhibitors for impact of CDK1 on TERT specifically? Basically did the authors try multiple CDK1 inhibitors and found some to be better than others for removing the phosphorylation event?

Reviewer #3 (Remarks to the Author): Expertise in telomerase

In the current manuscript, Yasukawa and colleagues define a role for CDK1 dependent phosphorylation of hTERT at T249 in cancer. It is proposed that this role is mediated via the function of hTERT as a RdRP.

The authors should indicate why they focused on pancreatic cancer and HCC in Figure 2 and better define the different stages of pancreatic cancer. In Supplemental Tables 2 and 3, it is not clear that T249-phosphorylation positive cases show higher incidence of lymph node metastasis than negative cases or that phosphorylation is associated with HCC grade.

The authors should provide more background on the previously reported function of hTERT as a RdRP and the assay first used in Figure 3 to monitor this activity.

Figure 3a, phosphatase treatment presumably affects all phosphorylation sites, not just T249, including the one at S824 that has been reported to regulate telomerase activity. Should an effect on telomerase activity be observed?

Figure 3d, levels of hTERT are definitely affected by CDK1 inhibition, suggesting in addition an indirect effect of reduced RdRP activity due to reduced levels of hTERT.

Figure 4a, using the PCR based telomerase assay, it may be important to assess a series of

dilution of the extracts in order to confirm activity is assessed in the linear range, and there are indeed no differences between the hTERT mutants and wild-type.

Figures 4b and 4c would be more convincing if the experiments were done with the BJ cells. In the control lanes, there should, using BJ cells lacking hTERT, be no telomerase activity and no RdRP activity.

In Figure 4d, the antibodies as characterized in supplemental Figure 1, should detect endogenous hTERT in the control lanes representing telomerase-positive cancer cell lines. Since these cell lines are telomerase-positive, are the authors suggesting that the effects of T249A on proliferation are due to dominant-negative effects?

The conclusion by the authors that the role of phosphorylation of T249 in hTERT in cancer cell proliferation is through activation of RdRP activity and is telomere-independent cannot be made. Experiments would have to be done in the context of cells lacking the telomerase RNA component in order for these conclusions to be drawn.

Supplementary Figure 7, the authors need to ascertain the interaction of hTERT with a different protein than nucleostemin, this is not one of the major telomerase-associated proteins.

Minor:

The last line of the abstract should possibly read '...in a telomere independent manner'.

Supplemental Figure 1a: What is the band at 102 kDa that also appears to be diminished upon sihTERT treatment?

The direct primer extension telomerase assays should include an internal control. In supplemental Figure 1b, a reaction from cells treated with sihTERT should be included.

Page 15, the expression of hTERT alleles does not clearly separate hTERT mediated telomerase and RdRP activity; the CRISPR experiments are important to remove endogenous hTERT, however, not because hTERT is expressed at low levels in cancer cells. Depending on the cancer cells, hTERT is expressed at significant levels, and is the limiting factor for telomerase activity.

Figure 6d, the authors indicate that they can't detect endogenous FOXO4 with the antibody they use, but in Figure 6 f, endogenous FOXO4 is detectable.

Reviewer #4 (Remarks to the Author): Expertise in FOXO regulation

The study by Yasukawa, identifies regulation of hTERT by CDK1 during mitosis and finds a non-canonical role for hTERT regulated by CDK1 in regulating expression of FOXO4.

This is an interesting and technically very well executed study where the experiments clearly support the conclusions drawn.

Phosphorylation of hTERT has been reported before but as mentioned by the authors in the discussion these previous reports of hTERT phosphorylation rely on the presence of 'consensus sites' and are not confirmed by an unbiased mass-spec approach. This conclusion is further confirmed when looking at the phosphosite.org website (no mass-spec data available on previous acclaimed phosphorylated-sites), unfortunately same accounts for the site identified in this study.

Irrespective the authors provide sufficient evidence that indeed this phosphosite is "real". The role of this phosphorylation is clearly not in regulating telomerase activity and the identification of FOXO4 as a result of mRNA regulation is interesting. They provide some evidence that this explains the role of CDK1/hTERT in cancer but this part may need some more experiments

(-) FOXO4 expression in contrast to FOXO1/3 is usually very low so whereas its expression may be increased due to loss of cdk1-mediated hTERT phosphorylation it is unclear when this may happen. CDK1 activity is low outside mitosis, but figure 1 shows that hTERT expression is low outside mitosis (or at least in non-nocadazole treated cells). So when is this regulation physiologically important ?

(-) Nocadazole treatment by itself induces JNK and p38 activation, so is the observed presence of hTERT in mitosis due to nocadazole treatment (through these stress kinases) or does hTERT expression cell cycle dependent and does the authors also observe increased hTERT expression in a non-perturbed cell cycle profile ?

(-) FOXOs are themselves substrates for CDKs and CDK1 in particular (although not (yet ?) reported for FOXO4). In case of FOXO1 it is unclear whether CDK1 (or CDK2) phosphorylation activates or inactivates FOXO1 transcriptional activity, so the verdict is still out (see e.g. Liu P, Kao TP, Huang H. *Oncogene*. 2008 Aug 7;27(34):4733-44. and in contrast Yuan Z, et al. *Science*. 2008 Mar 21;319(5870):1665-8. So the authors could/should investigate if and how CDK1 regulates FOXO4. As this may affect their conclusions regarding the role of FOXO4 in their observations.

(-) The above is of particular importance as FOXOs are not classical tumorsuppressors (haploinsufficient). Activity is not really depending on protein concentration but rather on their regulatory inputs (e.g. PI3K).

(-) Also recent reports indicate that FOXOs are depending on the context either oncogenic or tumorsuppressor (see for example Hornsveld et al. *Cancer Res*. 2018 May 1;78(9):2356-2369.). Thus it can not be concluded solely on literature how FOXO4 acts in the context that the authors study.

Detailed response to the Reviewers

Reviewer #1: Expertise in CDK1

“The study is technically sound and contains nice multiple-level proofs for the proposed model. However, there are few important points outlined below that should be addressed before publishing. These mostly concern the direct correlation of T249 phosphorylation with elevated CDK1 activity, and validation of glutamic acid as a true phospho-mimetic in this particular context.”

The reviewer stated that *“the study is technically sound and contains nice multiple-level proofs for the proposed model,”* but requested a few experiments to be addressed. We thank the reviewer for these suggestions and have addressed each of her/his points in detail below.

- 1) *“The authors should provide direct evidence of CDK1 phosphorylation effect on hTERT RdRP activity. While they showed that expression of the phosphorylation-defective mutant hTERT T249A decreased RdRP activity, and hTERT T249E mutant increased RdRP activity, suggesting that this mutant acts as a phosphomimetic mutant, they did not provide direct evidence that CDK1 phosphorylated T249 does the same in vitro. It would be easy to use the phosphatase to remove the phosphate groups from hTERT proteins, like in Figure 3A, and then get rid of the phosphatase and add CDK1 and ATP to restore the phosphorylation. And then, perform the assays again. This would support the claim that the T249E really acts as a true mimetic for CDK1 phosphorylation, and would allow to postulate with sufficient certainty, that phosphorylation of hTERT T249 is required for the RdRP activity but not for reverse transcriptase activity of hTERT.”*

In our original submission, we identified that CDK1 phosphorylates hTERT at threonine 249 (T249) in mitotic phase and the phosphorylation of T249 is necessary for RdRP activity (original Figure 3a, b and c). We demonstrated that T249 of the recombinant hTERT protein (191a.a.-306 a.a.) is phosphorylated by CDK1 *in vitro* (original Figure 1h) and the correlation between T249 phosphorylation of hTERT and RdRP activity *ex vivo* using cell lines (original Figure 3b and c); however, we agree that we did NOT provide direct evidence that phosphorylation of T249 by CDK1 directly regulates RdRP activity *in vitro*. As suggested by this reviewer, we have now performed an *in vitro* kinase assay. Specifically, we treated the hTERT immune complex with phosphatase to remove phosphate groups from hTERT protein (original Figure 3a) and then added CDK1 and ATP *in vitro* to phosphorylate T249 occurs in endogenous full-length hTERT. We are now able to provide *in vitro* direct evidence that CDK1 phosphorylates T249 (revised Figure 3b; left panel) and that this phosphorylation event regulates RdRP activity (revised Figure 3b; right panel). We incorporated these new experiments in the revised manuscript (revised Figure 3b) and discussed the point (revised manuscript page 13-14). We thank for this reviewer for this important and constructive suggestion, which we agree strengthens the manuscript. Please also see the response to reviewer #1 point 7.

- 2) *“The authors should provide proof that the used cancer models with higher RdRP activity, and individual cells immuno-stained for phospho-T249, really had higher CDK1 activity. They only state that previous reports have demonstrated that CDK1 expression is upregulated in many human malignant tumor tissues and that CDK1 activity correlates with the prognosis of patients with tumors. However, the study does not provide any direct experimental demonstration for such a correlation between CDK1 activity and phosphorylated T249 in tumor cells. They also claim that “the phosphorylation occurs more frequently in aggressive and advanced cancers in clinical samples”. Is this higher frequency correlated with higher CDK1 activity relatively to less advanced cancers?”*

In our original submission, we demonstrated that hTERT phosphorylation at T249 occurs in cancers, more frequently in advanced cancers in a series of immunohistochemical analyses (original Figure 2). More specifically, we observed that the number of 249T-P positive cells was the highest in carcinomas, followed by PanIN-3, PanIN-2, PanIN-1, and duct epithelium in pancreatic ductal lesions (original Figure 2i). Moreover, we found that 249T-P positive cases showed higher incidence of lymph node metastasis than 249T-P negative cases in pancreatic cancer (original Supplementary Table 2). In addition, we observed that patients whose tumors lacked 249T-P staining had longer overall survival and recurrence-free survival than patients whose cancers with 249T-P staining positive (original Figure 2n and 2o) in liver cancer. Therefore, we concluded that the phosphorylation of T249 occurs more frequently in aggressive and advanced cancers in clinical samples.

In addition to these our own observations, previous studies demonstrated that CDK1 expression is upregulated in many human malignant tumor tissues and that CDK1 activity correlates with the prognosis of patients with tumors (Nat Rev Cancer 2009, Malumbres *et al.*, Nat Rev Drug Discov. 2015, Asghar *et al.* and Gene 2019, Piao *et al.*), thus we speculated a link between CDK1 and phosphorylation of hTERT at T249 in clinical samples. However, we agree that we did NOT provide any direct experimental demonstration for such a correlation between CDK1 expression and phosphorylated T249 in tumor tissues as suggested by this reviewer. We therefore performed immunohistochemical analyses for CDK1 staining with the identical clinical samples for the phosphorylation of hTERT at T249. In addition, we performed fluorescent double staining with T249 phosphorylation and CDK1 to monitor the positive correlation in individual cells in clinical samples. We have now performed CDK1 staining in a series of pancreatic cancer and liver cancer tissues (revised Figures 2i, 2o and 2p). We found that, at 10% cutoff, clinicopathological analysis revealed strong correlation between CDK1 expression and phosphorylation of T249 of hTERT ($p=0.0289$ in pancreatic cancer and $p<0.0001$ in liver cancer, respectively) (revised Supplementary Table 2, 3 and 4). In addition, double staining with T249 phosphorylation and CDK1 in pancreatic cancer sample demonstrated that fluorescent intensity of T249 phosphorylation and CDK1 showed positive correlation at single cell level ($p=0.0350$, revised Supplementary Figure 7). Moreover, phosphorylation of hTERT at T249 as well as high CDK1 expression was detected among HCCs at BCLC stages A (less advanced), B (intermediate), and C (advanced) (revised Supplementary Table 4). Both phosphorylation of hTERT at T249 ($p=0.048$) and high CDK1 expression ($p=0.036$) correlated with poor overall survival with statistical significance (revised Figure 2 q and r).

Taken together, we believe that we are now able to provide a direct evidence to support a correlation between CDK1 expression and phosphorylated T249 in clinical samples. We have incorporated the new data and extensively revised the figure (revised Figure 2, revised Supplementary Figure 7 and revised Supplementary Table 2-4) and discussed these points in the revised manuscript (revised manuscript page 12-13).

- 3) *“The following statement on page 8 needs more explanation: “We noted that the sequence surrounding T249 is part of a motif that is frequently phosphorylated in mitosis and has been implicated as a target of the serine-threonine kinase cyclin-dependent kinase 1 (CDK1).” What does “frequently phosphorylated in mitosis” exactly mean?”*

The reviewer is correct that this description is confusing. We intended to describe that the sequence surrounding T249 contains [S/T]P motif that is implicated as a target site of CDK1 (PNAS 2008, Dephoure *et al.*). We have revised the text to clarify this point (revised manuscript page 9). We thank the reviewer for pointing this out.

- 4) *“On Figure 3B, a control blot for hTERT protein levels should be presented for each lane.”*

We have now included this important control and added the Figure (revised Figure 3c: middle panel).

- 5) *“It is not correct to claim on page 17, first paragraph, that, taken together, these observations confirmed that phosphothreonine 249 in hTERT is essential for cancer cell proliferation through activation of RdRP activity of hTERT. Although in Figure 5E, the effect on tumor volume in this particular xenograft model was quite impressive, indeed, the other experiments in the study do not allow such absolute generalization of essentiality of this particular phosphorylation event for cancer cell proliferation.”*

The reviewer is correct that we were not accurate. We have now changed the phrasing more carefully (the words “confirmed” and “essential” have been changed to “favor” and “important”, respectively) (revised manuscript page 20).

- 6) *“Please clarify how these two statements fit together: (1) Page 24 second paragraph. “Several laboratories have shown that hTERT is phosphorylated. Specifically, four phosphorylation sites have been reported.” (2) Later, on the same page. “These phosphorylation sites were identified from consensus sequences for specific kinases, and MS analysis was not conducted in previous reports.” Does this mean that, in fact, the previous works did not demonstrate the phosphorylation of hTERT, but rather just predicted based on consensus sequences? There must be more to it, but as it is, the whole paragraph leaves the reader into total confusion. Moreover, the authors themselves state that they were unable to detect these sites by MS analysis in this study with mitotic cells. Further on, the authors speculate that the amount and level of phosphorylation in these sites may be difficult to detect using MS analysis. This leaves an impression that we can neglect these reports on phosphorylation, as if there is no quantitative MS phosphoproteomics data, it is hard to prove that the predicted phosphorylation events matter. The authors should elaborate on the experiments in literature more to give a clear picture of the evidence for a reader.”*

We intended to explain that phosphorylation of hTERT has been reported before from several laboratories, but these previous reports of hTERT phosphorylation largely rely on the prediction from the presence of consensus sites (motifs) and were not confirmed by an unbiased mass-spec approach. However, since we were unable to identify these putative phosphorylation sites by the mass-spec approach in this study, our goal was to discuss the reason why we were unable to identify these sites. Nevertheless, we agree that this part is confusing. We have now revised the manuscript in a way to clarify these points. Specifically, 1) we have summarized the previous reports about phosphorylation of hTERT from other laboratories (J Cell Sci 2012, Chung *et al.*, JBC 1999, Kang *et al.*, MCB 2003, Haendeler *et al.* and JBC 2013, Jung *et al.*), 2) explained the fact that we were unable to detect these sites based on our unbiased mass-spec approach (revised Supplementary Table 6) and 3) discussed the reason why the discrepancy occurs (revised manuscript page 29-30). We thank the reviewer for pointing this out and agree that the alterations make the manuscript clearer.

- 7) *“In connection with the previous point: Phosphate Affinity SDS PAGE western blotting of the IP from nocodazole arrested cells (samples w/o phosphatase like in Figure 1c, for example) would help to see if there are several phosphorylation events, or just one, occurring in vivo.”*

We agree that Phosphate Affinity SDS PAGE western blotting using Phos-tag® Acrylamide gel is one of the ways to evaluate the phosphorylation in detail. However, we have alternative data to discuss this point instead of performing Phosphate Affinity SDS PAGE.

As shown in response to reviewer #1 point #1, we confirmed that T249 residue of hTERT is phosphorylated by CDK1 using *in vitro* kinase assay and clearly indicated that phosphorylation of T249 is essential for RdRP activity (revised Figure 3b). In addition, we note that CDK1 treatment does

NOT completely relocate the hTERT signal to the original position (revised Figure 3b, lower panel). This strongly indicates that there are other phosphorylation event(s) besides T249 phosphorylation by CDK1. While this observation is also interesting, the current study focuses on the T249 phosphorylation and RdRP activity and the further investigations about the other phosphorylation site(s) are out of the scope of this manuscript. However, in relation to the response to reviewer #1 point #6 and reviewer #3 point #3, we discussed and clarify these points in the revised manuscript (revised manuscript page 13-14 and page 29-30).

Minor points

- 1) *“Mentioning of serine 227 on page 7 and Figure 1 is out of context and confusing, as it was not detected in MS in Figure 1d. Later mentioning in the main text with the literature reference is sufficient.”*

We thank the reviewer for this suggestion. We fixed this point in our revised manuscript.

- 2) *“Page 25/26: “When we introduced substitutions of T249 (T249A and T249E) into endogenous hTERT, we found that these mutants did not affect telomerase activity at telomeres but instead affected RdRP activity and inhibited the tumorigenic potential of human cancer cell lines.” The inhibition claim should be only for T249A, not for T249E?”*

The reviewer is correct that we were not accurate. We thank the reviewer for pointing this out. We fixed the point in the revised manuscript (page 32).

Reviewer #2: Expertise in telomerase

“Overall the data are sound and interpreted carefully. That being said I have a few suggestions as outlined below that could improve the impact of the manuscript.”

The reviewer stated that *“overall the data are sound and interpreted carefully,”* and suggested a few constructive suggestions to strengthen the manuscript. Certainly, the suggestions from this reviewer are very constructive and make the manuscript clear and strong. We very much thank the reviewer for the review and have addressed each of her/his suggestions in detail below.

- 1) *“Does the CDK1 inhibitor (RO-3306) impact telomerase activity by TRAP assay or direct telomerase assay?”*

In our original submission, we demonstrated that suppression of CDK1 with siRNAs specific for CDK1 had no effect on telomerase activity (original Figure 3c). In addition to genetic manipulation of CDK1 expression, inhibition of CDK1 activity by a chemical compound is equally important evidence to present. We have performed this experiment and confirmed that the CDK1 inhibitor (RO-3306) has no effect on telomerase activity by direct telomerase assay. We have incorporated the data in revised Figure 3f (left panel).

- 2) *“Can the authors provide evidence in the BJ cells that overexpressed WT TERT has RdRP activity? Using the T249A and T249E cell lines already generated and performing the RdRP assay should be sufficient. I ask because I am curious if normal cells have the needed cofactors for this alternative function of hTERT or if this is something that is utilized solely by cancer cells. This is critical information. It would appear from the clinical samples that the authors think that the answer would be no, normal cells with OE WT TERT do not have RdRP activity because T249 is not readily phosphorylated and they show that this even is critical for RdRP activity in the manuscript. Thus this experiment is critical to provide evidence to the authors idea that advanced cancers more readily utilize this alternative function of TERT.”*

We agree that this is a very important point. As this reviewer suggested, we have now performed the RdRP assay using BJ cells stably expressing WT-hTERT, T249A and T249E. As the reviewer assumed, BJ cells stably overexpressing WT-hTERT and T249A cells exhibited little RdRP activity while T249E cells demonstrated steady state level of RdRP activity indicating that normal cells exhibit little RdRP activity due to the lack of phosphorylation at T249 of hTERT. Therefore, we confirmed that the phosphorylation at T249 of hTERT is critical event. We have now included the data in revised Supplementary Figure 8 and discussed the point (revised manuscript page 15-16).

- 3) *“Why did the authors choose CAGE seq to analyze expression differences in the CRISPR lines? By my understanding of CAGE seq this is used to map TSS of long-lived mRNAs. This could significantly impact the results. While FOXO4 seems to be a strong candidate, could the authors provide a better rationale as to why CAGE seq was chosen over other RNA seq methods?”*

As correctly pointed out by the reviewer, the uniqueness of CAGE is to map transcription starting sites (TSSs) at base-pair resolution, which enables us to identify cis-regulatory elements, such as promoters and enhancers, and to delineate transcriptional regulatory networks. At the same time, CAGE is very comparable to other gene expression profiling methods using total RNA extracts (containing long-lived mRNAs mainly), such as RNA-seq (Genome Res. Kawaji *et al.*, 2014). Essentially the difference

between CAGE and RNA-seq is only the captured region within mRNA molecules, that is, 5'-ends (CAGE) or arbitrary fragments of exons (RNA-seq). Hence, we simply chose CAGE to characterize the cells to study gene expression and to have better understanding of transcriptional regulation. To clarify what we elected to use CAGE, we have clarified this point in the revised manuscript (revised manuscript 21). We thank the reviewer for pointing this out.

- 4) *“The hTERT RNA IP assays could be improved by showing a direct interaction of hTERT with FOXO4A exon 1. If this interaction is direct the authors should be able to crosslink using UV and then IP with hTERT and measure Exon 1 FOXO4 RNAs. Have the authors tried this experiment? The washing in a non-crosslinked IP is not sufficient to rule out cell mixing issues during lysis.”*

The reviewer assumed that if the interaction between hTERT and FOXO4 RNA is direct, the association after UV crosslinking should be maintained even under the stringent washing condition such as high-salt condition (1M NaCl). In our original submission, we washed the immunocomplex with 300 mM NaCl in “a non-crosslinked IP” condition. As suggested by this reviewer, we have performed UV crosslinking followed by conventional immunoprecipitation with hTERT antibody and RT-PCR. In this experiment, we washed the immunocomplex with up to 1M NaCl condition and confirmed that interaction is maintained in such a stringent condition to rule out the non-specific association *in vitro*. We have now added the data (revised Supplementary Figure 15) and revised the manuscript (page 23). We thank the reviewer for this constructive suggestion.

- 5) *“Could the authors treat WT TERT cancer cells with the suspected anti-sense FOXO4 exon 1 RNA that would be generated from the TERT binding and producing RdRP antisense transcripts? This experiment would provide further evidence that this is indeed the mechanism of action. The overexpression and knockdown experiments are a nice start but the piece of data that is missing is the treatment of cells with FOXO4 antisense E1 transcripts that should also knockdown FOXO4 in the fashion the authors are proposing.”*

We have now performed the experiment as suggested by this reviewer. Specifically, to synthesize an antisense RNA complementary to the first exon of FOXO4 mRNA, a part of the first exon of FOXO4 mRNA was amplified by PCR and the PCR amplicon which has T7 promoter sequence was used as the template for *in vitro* transcription by T7 RNA polymerase. The synthesized antisense RNA was transfected in T249A-CRISPR cell and Control-CRISPR (WT TERT cancer cells) and determined whether FOXO4 mRNA is regulated in an antisense RNA dependent manner. As shown in the revised Figure 6d, we have confirmed that FOXO4 antisense exon 1 transcripts down regulate FOXO4 mRNA. We have now included the data (revised Figure 6d), discussed the points in revised manuscript (page 23) and the material and method are also described (page 55). We believe that the alterations significantly strengthen the manuscript. We very much appreciate the reviewer for this insightful and constructive suggestion.

- 6) *“Given that the 249T amino acid is not in the TERC RNA binding domain it makes sense that this residue is dispensable for TERC binding but did the authors confirm this in their CRISPR assays? I know the direct assay still worked and TRAP but this would confirm those findings and the authors should already have the IP cDNAs to do the assay. Using the IP samples the authors should confirm hTERC binding to mutant TERT.”*

We have now confirmed hTERC binds to endogenous WT-hTERT and T249A-CRISPR (mutant hTERT) equally as the reviewer predicted. We have now included the data in our revised manuscript (revised Figure 5f, lower panel).

- 7) “Second question along this line of thought is from the 2009 Nature paper by this same group *ALU RNAs were the second most abundant RNA found to IP with TERT. Does the mutant still bind ALUs? ALU transcripts are associated with gene expression regulation as well and could further explain the growth phenotypes observed.*”

While we indeed reported that Alu RNAs were the second most abundant RNA recovered by TERT immunoprecipitation (Nature 2009, Maida *et al.* Supplementary Table 1), at the same time we clearly described that we failed to confirm the interaction of Alu RNA with TERT by a biased RT-PCR with specific primer sets (shown in the figure) for Alu (Nature 2009, Maida *et al.* METHOD section). So, it is unlikely that Alu transcripts are associated with gene expression and involve in observed growth phenotype. Nevertheless, we re-confirmed that WT-hTERT nor T249A mutant hTERT does not interact with Alu RNA by confirming with the same primer sets in the Nature paper. We presented the data only in this detailed response because it is not within the scope of our study to investigate conditions of hTERT and Alu interaction.

- 8) “Do the authors speculate that in cells that appear to be addicted to TERT, that is die rapidly upon TERT inhibition by siRNAs, that is activity is responsible for those findings? It would appear to be the case.”

Although telomere maintenance by hTERT protects telomeres, several lines of evidence implicate TERT plays role(s) at locations distinct from telomeres. As this reviewer may be aware, in the history of our study about TERT- RdRP activity, we have been reporting that hTERT localizes to both mitotic spindles and centromeres and hTERT protein expression level is enriched in mitosis (MCB 2014, Maida *et al.* and MCB 2016, Maida *et al.*) since the first description of RdRP activity of hTERT from our laboratory (Nature 2009, Maida *et al.*). In addition to biochemical analysis of RdRP activity, we have reported several biological observations that implicate functional role(s) of hTERT beyond telomere maintenance (PNAS 2012, Okamoto *et al.*, MCB 2014, Maida *et al.* and MCB 2016, Maida *et al.*). As this reviewer correctly assumed, based on these previously published observations we have been speculating that hTERT inhibition by siRNAs induces “acute phenotype”.

We have now determined more precisely whether suppression of hTERT by siRNA may affect the cell viability using several different cancer cell lines known to be hTERT positive (A549 cells, 293T cells and HeLa cells) and an hTERT null cell line (VA13). As shown in Supplementary Figure 10, we have confirmed that cancer cells addicted to TERT rapidly die by suppression of hTERT. We have now included the data and discussed the point (revised manuscript page 18). In addition, we have conducted an experiment to determine whether over expression of phospho-inhibitory mutant (T249A-hTERT), phospho-mimetic mutant (T249E-hTERT) or WT-hTERT affects the cell proliferation in Saos2 cells (hTERT null cells). We observed that T249E mutant most effectively accelerates the cell proliferation, WT-hTERT the second and that T249A has no effects on cell proliferation. Since the data support our model, we have also included the data in revised Supplementary Figure 9. Please also see the response to the reviewer #3 point #8.

- 9) “CDK1 is upregulated in many cancer types. Pan-CDK inhibitors are in clinical trials. Can the author provide insights into using CDK1 inhibition in telomerase positive cancers and if there are better inhibitors for impact of CDK1 on TERT specifically? Basically did the authors

try multiple CDK1 inhibitors and found some to be better than others for removing the phosphorylation event?”

As the reviewer indicated, CDK1 is essential for progression into M-phase and is often overexpressed in human cancers, including HCC (Nature 2004, Massague *et al.*). In addition, a previous report indicated that CDK1 overexpression correlated with poor clinicopathological features and survival in HCC (Cancer Genet. 2017, Agarwal *et al.*) and this finding is consistent with our current study. Indeed, following the success of a pan-CDK inhibitor, palbociclib, on FDA approval in advanced breast cancer, two phase II clinical trials (milciclib and palbociclib) are currently ongoing to evaluate their efficiency on advanced HCC. Unfortunately, milciclib targets CDK2/4/5/7 and palbociclib targets CDK4/6, and no pan-CDK inhibitors that also targets CDK1 is currently evaluated in clinical trials (Hepatology Research 2019, Shen *et al.*).

Since hTERT is the most frequent driver genes activated in several cancers and our data demonstrated the novel role of CDK1 to phosphorylate hTERT T249 and induce tumorigenesis in a FOXO4 dependent manner, selective CDK1 inhibitors may have the profound impact on the treatment of aggressive hTERT positive cancers and pan-CDK inhibitor(s) that also targets CDK1 would be a better candidate for treating hTERT positive cancers, which may pave the way for the personalized treatment of cancer patients. Unfortunately, at the present time the commercially available CDK1 inhibitor RO3306 (PNAS 2006, Vassilev *et al.*) appears to be the best available inhibitor for CDK1. Nevertheless, we agree that it is very important to discuss this point and we have incorporated the point in our revised manuscript (revised manuscript page 31).

Reviewer #3: Expertise in telomerase

- 1) *“The authors should indicate why they focused on pancreatic cancer and HCC in Figure 2 and better define the different stages of pancreatic cancer. In Supplemental Tables 2 and 3, it is not clear that T249-phosphorylation positive cases show higher incidence of lymph node metastasis than negative cases or that phosphorylation is associated with HCC grade.”*

We focused on the characterization of T249 phosphorylation in pancreatic cancer and HCC, since pancreatic cancer and HCC are most deadly cancers with poor survival outcome (CA Cancer J Clin. 2018, Bray *et al.*). In addition, TERT activation plays a fundamental role in the development of these cancers and SNPs at TERT are reported to correlate with increased risk of pancreatic cancer (Gut 2017, Bao *et al.*). TERT is a key driver gene activated in HCC, and its promoter mutations were most common somatic mutations found in HCC (Cell 2017 169(7): 1327-1341, Cancer Genome Atlas Research Network). In conclusion, pancreatic cancer and HCC are two major clinically important most deadly cancers and both cancers probably occur in correlation with functional role of hTERT protein. We clarify and incorporated the point in the revised manuscript (page 28). In addition, as for the “better define of the different stages of pancreatic cancer”, we added the detailed description of different stages of pancreatic cancer in the material and method section (revised manuscript page 51-52).

As for the “incidence of lymph node metastasis” and “the phosphorylation and HCC grade”, in our original submission, we clearly observed the significant difference in p value based on the statistically analysis and demonstrated the data in the Tables (original Supplementary Table 2 and 3). To clearly present the difference at a glance, we have graphically demonstrated that lymph nodes metastasis was more frequently detected in pancreatic cancer with T249 phosphorylation (revised Supplementary Figure 6, panel a). We have also graphically demonstrated that T249 phosphorylation correlated with the histological degree of differentiation in HCC (revised Supplementary Figure 6, panel b).

- 2) *“The authors should provided more background on the previously reported function of hTERT as a RdRP and the assay first used in Figure 3 to monitor this activity.”*

We thank the reviewer for this suggestion to give us an opportunity to explain the point. We have now expanded the manuscript in both introduction section and results section (revised manuscript page 4 and page 7; the first description of IP-RdRP assay in this manuscript) and clarify the point. Specifically, we have described the points: 1) we and others observed that hTERT expression is the highest in mitotic phase (NAR 2014, Xi/Cech, MCB 2014, Maida *et al.* and MCB 2016, Maida *et al.*), 2) we reported that mitotic hTERT has an RdRP activity (MCB 2016 Maida *et al.*) and that is the reason why we speculated the link between an important mitotic kinase, CDK1, and RdRP activity of hTERT in mitosis, and 3) the detailed explanation of how to establish the RdRP assay from the first detection of hTERT-RdRP activity in nuclear extracts derived from HeLa cells arrested in mitosis (MCB 2014, Maida *et al.*) and further confirmation of these observations (MCB 2016, Maida *et al.* and JoVE 2018, Maida *et al.*).

- 3) *“Figure 3a, phosphatase treatment presumably affects all phosphorylation sites, not just T249, including the one at S824 that has been reported to regulate telomerase activity. Should an effect on telomerase activity be observed?”*

In our original submission, we demonstrated that phosphatase treatment had no effect on telomerase activity (original Figure 3a). On the other hand, there was a report that indicated that S824 of hTERT was phosphorylated by Akt kinase and that phosphorylation at S842 of hTERT enhanced telomerase

activity (JBC 1999, Kang *et al.*). Since the phosphatase treatment removes not only the phosphate group at T249 but also other phosphate group such as S824 (if any) of hTERT, the reviewer assumes that the phosphatase treatment must/should have effect on telomerase activity.

The reviewer is completely correct and we agree that “phosphatase treatment presumably affects all phosphorylation sites, not just T249”, however, as written in our original submission, we were unable to detect phosphorylation at S824 previously reported by a single group (JBC 1999, Kang *et al.*) by our unbiased mass-spec approach. In addition, the study from this group did not provide any direct evidence that hTERT S824 phosphorylation itself is really essential and prerequisite for the maintenance of basal telomerase activity. Indeed, in their original paper by Kang (JBC 1999, Kang *et al.*), one could identify telomerase activity even after treatment with phosphatase which is comparable to the basal level of telomerase activity without any treatment. We speculated that although hTERT S824 phosphorylation (if any) may enhance the telomerase activity, phosphatase treatment to remove phosphate group(s) in hTERT did not affect the minimal basal activity of telomerase, especially at the condition without artificially induced Akt kinase activity. Nevertheless, we now discuss this point in the revised manuscript (revised manuscript page 29-30).

- 4) *“Figure 3d, levels of hTERT are definitely affected by CDK1 inhibition, suggesting in addition an indirect effect of reduced RdRP activity due to reduced levels of hTERT.”*

We agree that levels of hTERT appeared to be slightly decreased (original Figure 3d: lower panel) by treatment with CDK1 inhibitor in the experiment that was shown. We have now repeated the same experiment in multiple times. In the end, we have now concluded that the levels of hTERT are the same even after the treatment with CDK1 inhibitor. So, it is unlikely that CDK1 inhibitor reduces the RdRP activity by an indirect effect of hTERT reduction. We have now revised the figure (revised Figure 3e).

- 5) *“Figure 4a, using the PCR based telomerase assay, it may be important to assess a series of dilution of the extracts in order to confirm activity is assessed in the linear range, and there are indeed no differences between the hTERT mutants and wild-type.”*

The reviewer is correct. We have now confirmed there are no differences between the hTERT mutants and wild-type in a serial dilution of the extracts from the TRAP assay and have added this experiment (revised Supplementary Figure 8a).

- 6) *“Figures 4b and 4c would be more convincing if the experiments were done with the BJ cells. In the control lanes, there should, using BJ cells lacking hTERT, be no telomerase activity and no RdRP activity.”*

We have now performed these experiments and added the data (revised Supplementary Figure 8b and 8c). Please also see the response to the reviewer #2 point #2.

- 7) *“In Figure 4d, the antibodies as characterized in supplemental Figure 1, should detect endogenous hTERT in the control lanes representing telomerase-positive cancer cell lines. Since these cell lines are telomerase-positive, are the authors suggesting that the effects of T249A on proliferation are due to dominant-negative effects?”*

Several laboratories have found that endogenous hTERT is not abundant and it is difficult to detect endogenous hTERT without immunoprecipitation with anti-hTERT antibody (NAR 2014, Xi/Cech). We completely agree that it is difficult to detect endogenous hTERT by western blotting. Therefore, we usually immunoprecipitate hTERT proteins with anti-hTERT antibody followed by western

blotting to detect “endogenous” hTERT (original Figure 1a, 1b, 1c, 1f, 1g, 3b, 3d and Supplementary Figure 1). In our manuscript, original Figure 4d does NOT depict “endogenous” hTERT- these cells stably overexpress control-empty vector, T249A mutant or T249E mutant. We have clarified and revised this point in the main text (page 17) and the Figure legend (revised Figure legend to Figure 4).

- 8) *“The conclusion by the authors that the role of phosphorylation of T249 in hTERT in cancer cell proliferation is through activation of RdRP activity and is telomere-independent cannot be made. Experiments would have to be done in the context of cells lacking the telomerase RNA component in order for these conclusions to be drawn.”*

This reviewer is correct. We agree that we need to do the experiment using cells lacking telomerase RNA component or telomerase activity to draw such a conclusion. We have conducted an experiment to determine whether over expression of phospho-inhibitory mutant (T249A-hTERT), phospho-mimetic mutant (T249E-hTERT) or WT-hTERT may affect the cell proliferation in Saos2 cells that lack telomerase activity. We observed that T249E mutant the most effectively accelerates the cell proliferation, WT-hTERT is the second and that T249A has no effects on cell proliferation (revised Supplementary Figure 9). While these data indicate that phosphorylation at T249 of hTERT is important for cell proliferation and might suggest that the effect is through RdRP activity, we agree that these experiments are not conclusive. We have added the data in revised manuscript and have mentioned these caveats.

- 9) *“Supplementary Figure 7, the authors need to ascertain the interaction of hTERT with a different protein than nucleostemin, this is not one of the major telomerase-associated proteins.”*

In our original submission, we ascertained that introduction of alanine or glutamic acid substitution at T249 in hTERT by CRISPR-Cas9 genome editing had no effect on protein-protein interactions, such as between hTERT and nucleostemin (NS). In previous studies, several labs reported that human hSMG-6 (also called hEST1) interacts with hTERT by assessing telomerase activity (Current Biology 2003, Reichenbach *et al.*, Snow *et al.* and NAR 2007, Redon *et al.*). In addition to nucleostemin (original Supplementary Figure 7), we have now demonstrated a direct interaction between hTERT and hSMG-6 and confirmed that introduction of these mutations into hTERT had no effect on these protein-protein interactions (revised Supplementary Figure 12).

Minor:

- 1) *“The last line of the abstract should possibly read ‘.in a telomere independent manner’.”*

We fixed this point.

- 2) *“Supplemental Figure 1a: What is the band at 102 kDa that also appears to be diminished upon sihTERT treatment?”*

Since we have been using immunoprecipitation (IP) (10E9-2 mAb) followed by immunoblotting (IB) (2E4-2 mAb) to detect endogenous hTERT throughout the manuscript, we have focused on analyzing the band at 102 kDa on the left-handed panel (10E9-2 for IP and 2E4-2 for IB) of the original Supplementary Figure 1a. We have performed an experiment to address this reviewer’s question. As shown in the revised Supplementary Figure 1d, we were able to detect the band at 102 kDa without a reaction with primary antibody in the immunoblotting process and this result indicates that the signal was nonspecific signal from the secondary antibody (revised Supplementary Figure 1d: “No primary Abs”). In addition, Mouse IgG as an irrelevant monoclonal Ab for the immunoprecipitation process

also recovers this 102kDa band (revised Supplementary Figure 1d: “Mouse IgG” lane) while “Beads only” failed to recover the band. Taken together, the band at 102 kDa is from a protein that is “nonspecifically” immunoprecipitated by a mouse IgG and “nonspecifically” detected by secondary antibody (anti-mouse IgG from rat) in the IB. Since the field (telomerase) has been suffered from lacking reliable antibody to detect endogenous hTERT, this information is very important and helpful in the future, we have included these data in our revised manuscript (revised Supplementary Figure 1d).

In addition, we agree that the band at 102 kDa appear to be diminished upon si-hTERT treatment especially in the right-handed panel (abx120550 for IP and ab32020 for IB) of original Supplementary Figure 1a. However, as indicated above and we used these combinations of antibodies (abx120550 and ab32020) to accurately and extensively validate our house-made antibodies (10E9-2 and 2E4-2) in this manuscript in side-by-side experiment, characterization of abx120550 or ab32020 is out of the scope of the manuscript.

- 3) *“The direct primer extension telomerase assays should include an internal control. In supplemental Figure 1b, a reaction from cells treated with sihTERT should be included.”*

Since the direct telomerase assay does not involve PCR, the assay usually does not include an “internal control” but instead includes a “recovery/loading standard” (Science 2007, Cohen *et al.*) to eliminate the possibility of uneven sampling loss during the assay. To avoid a misevaluation by such a handling error, we usually performed the direct telomerase assay in technical-triplicates (original Supplementary Figure 1b) instead of including “recovery/loading standard”. We do not believe that we lose the samples in the middle of assay and misevaluate the telomerase activity in our original submission. However, we demonstrated to include the “recovery/loading standard” in a series of revised experiments (revised Supplementary Figure 1c and 8b). In addition, we performed the direct telomerase activity assay from cells treated with si-hTERT as suggested by this reviewer (revised Supplementary Figure 1c). We thank the reviewer for this suggestion.

- 4) *“Page 15, the expression of hTERT alleles does not clearly separate hTERT mediated telomerase and RdRP activity; the CRISPR experiments are important to remove endogenous hTERT, however, not because hTERT is expressed at low levels in cancer cells. Depending on the cancer cells, hTERT is expressed at significant levels, and is the limiting factor for telomerase activity.”*

The reviewer is completely correct and we were not accurate. We have revised the manuscript (revised manuscript page 18).

- 5) *“Figure 6d, the authors indicate that they can’t detect endogenous FOXO4 with the antibody they use, but in Figure 6f, endogenous FOXO4 is detectable.”*

It is true that we were unable to detect endogenous FOXO4 with the antibody that we used as demonstrated in original Figure 6b (the 1st lane), 6d (the 1st lane) and original Supplementary Figure 10a (the 1st lane). As the reviewer indicated, in contrast, we detected FOXO4 in original Figure 6f (the 1st lane). Since the cells presented in Figure 6f (in the 1st lane) is T249A CRISPR cells which over express FOXO4 by modulation of T249 residue with CRISPR/Cas9 genome editing, we are able to detect endogenous (upregulated) FOXO4. (The 1st lane of the original Figure 6f corresponds to the 2nd lane of the original Figure 6b.) To clarify this point, we added explanation to the Figure legend (revised Figure legend to revised Figure 6b and 6e).

Reviewer #4: Expertise in FOXO regulation

This is an interesting and technically very well executed study where the experiments clearly support the conclusions drawn.

Phosphorylation of hTERT has been reported before but as mentioned by the authors in the discussion these previous reports of hTERT phosphorylation rely on the presence of 'consensus sites' and are not confirmed by an unbiased mass-spec approach. This conclusion is further confirmed when looking at the phosphosite.org website (no mass-spec data available on previous acclaimed phosphorylated-sites), unfortunately same accounts for the site identified in this study. Irrespective the authors provide sufficient evidence that indeed this phosphosite is "real".

The role of this phosphorylation is clearly not in regulating telomerase activity and the identification of FOXO4 as a result of mRNA regulation is interesting. They provide some evidence that this explains the role of CDK1/hTERT in cancer but this part may need some more experiments

The reviewer stated that *“This is an interesting and technically very well executed study where the experiments clearly support the conclusions drawn.”* We very much appreciate the positive and supportive evaluation by this reviewer. In addition, the reviewer suggested several insightful and constructive suggestions to strengthen the manuscript. We thank the reviewer for these important comments and have addressed each of her/his points in detail below.

- 1) *“(-) FOXO4 expression in contrast to FOXO1/3 is usually very low so whereas its expression may be increased due to loss of cdk1-mediated hTERT phosphorylation it is unclear when this may happen. CDK1 activity is low outside mitosis, but figure 1 shows that hTERT expression is low outside mitosis (or at least in non-nocadazole treated cells). So when is this regulation physiologically important?”*

In our original submission, we identified novel phosphorylation site of hTERT (T249) and this phosphorylation occurs in mitotic phase by a mitotic kinase, CDK1. In addition, we demonstrated that the phosphorylation at T249 is important for hTERT RdRP activity in mitosis. Furthermore, we identified FOXO4 as a target gene to be downregulated by antisense RNA produced by RdRP activity of hTERT against FOXO4 transcripts. Since CDK1 activity is high in mitosis and RdRP activity of hTERT is also enriched in mitosis as well, the reviewer assumed that one may have a question that FOXO4 expression level is lower in mitosis than in other cell cycle phases. To address this reviewer's question, we monitored FOXO4 mRNA level by manipulating cells in mitosis. We have now confirmed that in mitotic phase FOXO4 expression level is lower than in other phases in WT-hTERT cells (Control-CRISPR cells) while we failed to observe the differences in T249A-CRISPR cells which defects in RdRP activity. This finding indicates that CDK1 activity and upregulated hTERT-RdRP activity effectively down-regulates FOXO4 expression in mitosis. We have included these data in the revised Supplementary Figure 13 and discussed the point in our revised manuscript (page 21-22).

- 2) *“(-) Nocadazole treatment by itself induces JNK and p38 activation, so is the observed presence of hTERT in mitosis due to nocadazole treatment (through these stress kinases) or does is hTERT expression cell cycle dependent and does the authors also observe increased hTERT expression in a non-perturbed cell cycle profile?”*

We have now demonstrated the experiment to manipulate cells in mitotic phase with nocodazole treatment or double thymidine block only (without nocodazole treatment). In both cases we observed the increase hTERT expression in mitotic phase. Consistent with our data, Cech laboratory also reported that hTERT expression is enriched in mitotic phase only by double thymidine block treatment

and release (NAR 2014, Xi/Cech). These data indicate that upregulation of hTERT is not due to nocodazole treatment (through stress kinases such as JNK and p38) but due to mitotic entry. We have incorporated the data (revised Supplementary Figure 2c) and explained in the revised manuscript (page 7).

- 3) “(-) FOXOs are themselves substrates for CDKs and CDK1 in particular (although not (yet ?) reported for FOXO4). In case of FOXO1 it is unclear whether CDK1 (or CDK2) phosphorylation activates or inactivates FOXO1 transcriptional activity, so the verdict is still out (see e.g. Liu P, Kao TP, Huang H. *Oncogene*. 2008 Aug 7;27(34):4733-44. and in contrast Yuan Z, *et al. Science*. 2008 Mar 21;319(5870):1665-8. So the authors could/should investigate if and how CDK1 regulates FOXO4. As this may affect their conclusions regarding the role of FOXO4 in their observations.”

We thank the reviewer for this comment. We have addressed to this question using T249A-CRISPR cells which exhibit increased FOXO4 expression due to loss of hTERT phosphorylation. When we treated the cell with CDK1 inhibitor (RO3306), we were unable to detect any differences of FOXO4 protein level and the migration of the FOXO4 protein. These data indicate that CDK1 does NOT involve in FOXO4 protein regulation by a direct phosphorylation of FOXO4 protein while we still cannot eliminate the possibility that FOXO4 phosphorylation (even if this occurs) does NOT affect protein migration in SDS-PAGE. We have included this data (revised Supplementary Figure 16 and page 23-24), cited all of the references suggested by the reviewer and discussed the point in detail (revised manuscript page 32-33).

- 4) “(-) The above is of particular importance as FOXOs are not classical tumorsuppressors (haploinsufficient). Activity is not really depending on protein concentration but rather on their regulatory inputs (e.g. PI3K).”

We have expanded the discussion about the point in the manuscript (revised manuscript page 32-33). Please also see the response to the reviewer #4 point #3.

- 5) “Also recent reports indicate that FOXOs are depending on the context either oncogenic or tumorsuppressor (see for example Hornsveld *et al. Cancer Res*. 2018 May 1;78(9):2356-2369.). Thus it can not be concluded solely on literature how FOXO4 acts in the context that the authors study.”

We very much appreciate this reviewer’s insightful comments. We have now cited all of the references suggested by this reviewer and have altered the discussion as the reviewer suggested. Please also see the response to the reviewer #4 point #3.

REVIEWERS' COMMENTS:

Reviewer #1 (Remarks to the Author):

The authors have significantly improved the manuscript and answered sufficiently to the raised criticism with a number of new experiments and improvements in the text. They now show direct evidence that CDK1 phosphorylates T249, and that this phosphorylation event regulates RdRP activity. They also provide evidence to support a correlation between CDK1 expression and phosphorylated T249 in clinical samples. I recommend to accept the manuscript for publishing.

Reviewer #2 (Remarks to the Author):

The authors report that telomerase RdRP activity in cancer cells is stimulated by phosphorylation at T249. In conjunction the phosphorylation event is critical in regulating the production of short antisense RNAs that regulate gene expression. The authors demonstrate that a key gene (FOXO4) is regulated in cancers by this unique activity of telomerase. This work represents a set of novel insights into non-canonical telomerase activities and provides a rationale as to why some cancer cells appear "addicted" to telomerase. While there is discrepancy in the field as to whether or not TERT/telomerase has other functions besides maintaining telomeres, the evidence presented here is sound. The cell line specificity of telomerase off-telomere functions is a topic for a different debate. The authors also provide substantial antibody validation which will be helpful to the telomere field in general. I think the authors have responded to my comments and provided substantial additional data and reached reasonable conclusions.

Reviewer #3 (Remarks to the Author):

My comments and concerns have been adequately addressed. The authors should include quantification of the TRAP assay products in Supplementary Figure 8A to confirm there are no differences in telomerase activity levels.

Reviewer #4 (Remarks to the Author):

The authors have addressed the issues I raised.

(-) I do find it not very reassuring that the phosphorylation site, which is the linchpin of this study cannot be identified by mass-spectrometry. Maybe years back one could argue that phosphoproteomics was a not yet mature research field technology wise, but that is no more.

(-)The question as to when this happens in real biology is also not really answers. FOXO4 detection is problematic (recognised also by other reviewer) and as indicated initially by me FOXO expression levels are hardly determining their biology.

(-) Finally in their answering on FOXO4 expression they refer to mRNA expression, mRNA expression globally is not predictive for protein expression so this cannot be translated 1:1 they should be aware of this

This manuscript has now seen so many reviewers that I am fine with publication, it's not going to be any different with additional questions and answers

Detailed response to the Reviewers

Reviewer #1:

“The authors have significantly improved the manuscript and answered sufficiently to the raised criticism with a number of new experiments and improvements in the text. They now show direct evidence that CDK1 phosphorylates T249, and that this phosphorylation event regulates RdRP activity. They also provide evidence to support a correlation between CDK1 expression and phosphorylated T249 in clinical samples. I recommend to accept the manuscript for publishing.”

Reviewer #2:

“The authors report that telomerase RdRP activity in cancer cells is stimulated by phosphorylation at T249. In conjunction the phosphorylation event is critical in regulating the production of short antisense RNAs that regulate gene expression. The authors demonstrate that a key gene (FOXO4) is regulated in cancers by this unique activity of telomerase. This work represents a set of novel insights into non-canonical telomerase activities and provides a rationale as to why some cancer cells appear "addicted" to telomerase. While there is discrepancy in the field as to whether or not TERT/telomerase has other functions besides maintaining telomeres, the evidence presented here is sound. The cell line specificity of telomerase off-telomere functions is a topic for a different debate. The authors also provide substantial antibody validation which will be helpful to the telomere field in general. I think the authors have responded to my comments and provided substantial additional data and reached reasonable conclusions.”

We thank the reviewers for their recommendation for publication.

Reviewer #3:

“My comments and concerns have been adequately addressed. The authors should include quantification of the TRAP assay products in Supplementary Figure 8A to confirm there are no differences in telomerase activity levels.”

We have quantified the products of TRAP assay in Supplementary Figure 8a. In addition to the result from direct telomerase assay (Supplementary Figure 8b), we have now confirmed that there are no differences in telomerase activity level. We incorporated these data in the final version of the Supplementary Figure 8a and 8b. We thank the reviewer for her/his recommendation for publication.

Reviewer #4:

The authors have addressed the issues I raised.

- 1) *“(-) I do find it not very reassuring that the phosphorylation site, which is the linchpin of this study cannot be identified by mass-spectrometry. Maybe years back one could argue that phosphoproteomics was a not yet mature research field technology wise, but that is no more.”*

We very much appreciate this reviewer for her/his continued and constructive suggestions to strengthen our manuscript.

In our current study, we identified T249 residue of hTERT as a novel phosphorylation site by mass-spectrometry approach. Other laboratories reported several putative phosphorylation sites different from the T249 residue (J Cell Sci 2012, Chung *et al.*, JBC 1999, Kang *et al.*, MCB 2003, Haendeler *et al.* and JBC 2013, Jung *et al.*) but we were unable to detect these previously reported sites by mass-spectrometry. We speculated that the experimental conditions where others conducted in their previous reports (such as under the overexpression of Akt kinase or under oxidative stress) were completely different from our experimental condition (mitotic phase). We further clarified this point in the final version of our manuscript (page 30).

- 2) *“(-)The question as to when this happens in real biology is also not really answers. FOXO4 detection is problematic (recognised also by other reviewer) and as indicated initially by me FOXO expression levels are hardly determining their biology.”*

We agree that this is important point. We have now further revised the manuscript in the final version of our manuscript in a way to clarify this point and tone down the conclusion (page 21-22).

- 3) *“(-) Finally in their answering on FOXO4 expression they refer to mRNA expression, mRNA expression globally is not predictive for protein expression so this cannot be translated 1:1 they should be aware of this.”*

We agree that this is also an important point. We have incorporated and clarified this point in the discussion section (page 33). We thank the reviewer for pointing this out.

- 4) *“This manuscript has now seen so many reviewers that I am fine with publication, it's not going to be any different with additional questions and answers.”*

We thank the reviewer for her/his recommendation for publication.